# Balancing Efficiency and Expressiveness: Subgraph GNNs with Walk-Based Centrality

## Abstract

We propose an expressive and efficient approach that combines the strengths of two prominent extensions of Graph Neural Networks (GNNs): Subgraph GNNs and Structural Encodings (SEs). Our approach leverages walk-based centrality measures, both as a powerful form of SE and also as a subgraph selection strategy for Subgraph GNNs. By drawing a connection to perturbation analysis, we highlight the effectiveness of centrality-based sampling, and show it significantly reduces the computational burden associated with Subgraph GNNs. Further, we combine our efficient Subgraph GNN with SEs derived from the calculated centrality and demonstrate this hybrid approach, dubbed HyMN, gains in discriminative power. HyMN effectively addresses the expressiveness limitations of Message Passing Neural Networks (MPNNs) while mitigating the computational costs of Subgraph GNNs. Through a series of experiments on synthetic and real-world tasks, we show it outperforms other subgraph sampling approaches while being competitive with full-bag Subgraph GNNs and other state-of-the-art approaches with a notably reduced runtime.

## 1 Introduction

Graph Neural Networks (GNNs) (Scarselli et al., 2009; Gori et al., 2005; Micheli, 2009) have achieved great success in learning tasks with graph-structured data. Typically, GNNs are based on the message passing paradigm (Gilmer et al., 2017), in which node features are aggregated over their local neighborhood recursively, resulting in architectures known as Message Passing Neural Networks (MPNNs). MPNNs have been shown to suffer from limited expressive power: they are bounded by 1-WL (Xu et al., 2018; Geerts & Reutter, 2023; Weisfeiler & Leman, 1968; Morris et al., 2019; 2023) cannot count certain substructures (Chen et al., 2020; Bouritsas et al., 2022; Tahmasebi et al., 2020) or solve graph bi-connectivity tasks (Zhang et al., 2023b).

In order to improve expressive power, some works augment the node features with Structural Encodings (SE) or Positional Encodings (PE) (Bouritsas et al., 2022; Dwivedi et al., 2021; Fesser & Weber, 2024). These predefined features provide additional information on the graph topology, leading to an improvement on benchmarks compared to using the raw features (Rampášek et al., 2022). Recently, Subgraph GNNs (Bevilacqua et al., 2021; Frasca et al., 2022; Zhang & Li, 2021; Cotta et al., 2021) have been proposed to overcome some of the expressivity limitations of MPNNs. This approach preserves equivariance while relying less on feature engineering. First, a Subgraph GNN transforms a graph into a "bag of subgraphs" based on a specific selection policy. These subgraphs are then processed by an equivariant architecture and aggregated to make graph- or node-level predictions. One common approach to generate subgraphs, known as node-marking, is to mark a single node in the graph (Papp & Wattenhofer, 2022). In this case, each subgraph is then "tied" to a specific node in the original graph, and a shared MPNN generates a representation for each subgraph.

In general, Subgraph GNNs show good empirical performance but come with a high computational cost. For a graph of $N$ nodes and a maximum degree $d_{\max}$, then Subgraph GNNs with node-marking based policies have a computational complexity $O(N^2 \cdot d_{\max})$. To reduce computational complexity, it was suggested sampling subgraphs randomly (Bevilacqua et al., 2021; Zhao et al., 2022) or, more recently, learning sampling policies (Qian et al., 2022; Bevilacqua et al., 2024; Kong et al., 2024). However, while random approaches have been shown to be suboptimal (Bevilacqua et al., 2024; Bar-Shalom et al., 2024b), learnable policies are difficult to train in practice and become impractical when sampling a larger number of subgraphs, due to the use of discrete sampling and RL-based objectives.

In this work, we present an approach that drastically reduces the complexity of Subgraph GNNs while maintaining their competitive performance. We achieve this by identifying easy-to-compute structural features that unlock a parsimonious yet effective subgraph selection strategy, while providing complementary information enhancing the model's expressiveness.

We first identify a family of node centrality measures (Estrada & Rodriguez-Velazquez, 2005; Benzi & Klymko, 2013; 2014) as easy-to-compute scores that: (i) recapitulate well the extent to which a subgraph alters the graph representation; (ii) correlate with relevant substructure counts. In light of this, we propose to leverage these measures to efficiently and effectively reduce the size of bags of subgraphs without requiring learning of additional components. Specifically, we prioritize selecting marked subgraphs associated with the top-ranking nodes according to these centrality measures and, in particular, to the Subgraph Centrality by Estrada & Rodriguez-Velazquez (2005).

Then, we propose to additionally interpret Centrality scores as Structural Encodings (CSEs), and to utilize them to augment the node features of the selected subgraphs. This combination of approaches is justified from an expressiveness perspective, as we show neither of the two subsume the other. We demonstrate node marking of (selected) subgraphs separates graphs that are not separable by CSE-based feature augmentations *and*, vice-versa, that CSEs allow to separate pairs indistinguishable to our subsampled Subgraph GNNs.

The resulting method, dubbed HyMN (*Hybrid Marking Network*), is an approach whereby Subgraph GNNs and Structural Encodings work in tandem to effectively overcome the expressiveness limitations of MPNNs and unlock the application of Subgraph GNNs to larger graphs previously out of reach. Our approach is provably expressive, does not require feature engineering or learnable sampling components and, importantly, maintains a low computational cost. The practical value of our approach is confirmed by the positive results achieved over a series of experimental analyses conducted over synthetic and real-world benchmarks. We show HyMN outperforms other subgraph selection strategies, and performs on par or better than full-bag Subgraph GNNs by only sampling one or two subgraphs. Additionally, HyMN is competitive to (and sometimes better than) Graph Transformers and other state-of-the-art GNNs, while, as we show through extensive wall-clock timing analyses, featuring a more contained computational run-time.

Our **contributions** are summarized as follows:

1. We show that walk-based node centrality measures and, in particular, the Subgraph Centrality of Estrada & Rodriguez-Velazquez (2005), represent a simple and effective indicator of subgraph importance for subsampling bags in Subgraph GNNs.
2. We demonstrate that centrality-based structural features can be employed as Structural Encodings to enhance the discriminative power of Subgraph GNNs with subsampled bags of subgraphs.
3. We provide strong experimental evidence showcasing the effectiveness of our sampling strategy and the efficacy of additionally incorporating centrality-based SEs.

Overall, our results validate our method as a simple, expressive and efficient approach that competes with state-of-the-art GNNs with reduced empirirical run-times.

## 2 PRELIMINARIES AND RELATED WORK

**Expressive power of MPNNs**. The expressive power of MPNNs has become a central research topic since they were shown to be bounded by the 1-WL isomorphism test (Morris et al., 2019; Xu et al., 2018; Morris et al., 2023). This has led to approaches which aim to obtain different representations for non-isomorphic but 1-WL equivalent graphs. These include using random features (Sato et al., 2021; Abboud et al., 2020), higher-order message passing schemes (Bodnar et al., 2021b;a; Morris et al., 2019; 2020b) and equivariant models (Maron et al., 2018; Vignac et al., 2020). One of the most common approaches is to inject positional and structural encodings into the input layer (Bouritsas et al., 2022; Fesser & Weber, 2024; Dwivedi et al., 2021; Kreuzer et al., 2021) using Laplacian PEs (Dwivedi & Bresson, 2020; Kreuzer et al., 2021; Wang et al., 2022; Lim et al., 2023; 2024) or distance information (Ying et al., 2021; Li et al., 2020). A random-walk based encoding (RWSE) was proposed in (Dwivedi et al., 2021) and was shown to be able to distinguish regular graphs such as the 1-WL-equivalent CSL graphs (Murphy et al., 2019).

**Subgraph GNNs.** A recent line of work has proposed representing a graph as a collection of subgraphs obtained by a specific selection policy (Zhang & Li, 2021; Cotta et al., 2021; Papp & Wattenhofer, 2022; Bevilacqua et al., 2021; Zhao et al., 2022; Papp et al., 2021; Frasca et al., 2022; Qian et al., 2022; Huang et al., 2023; Zhang et al., 2023a). These approaches, jointly referred to as Subgraph GNNs, allow to overcome the expressivity limitations of MPNNs without introducing predefined encodings. A powerful, common selection policy termed *node marking* involves generating a subgraph per node by marking that node in the original graph, with no connectivity alterations. Although it can increase expressive power beyond 1-WL (Frasca et al., 2022; You et al., 2021), this approach has a high computational complexity as we need to consider and process $N$ different subgraphs, where $N$ is the number of nodes in the original graph.

Several recent papers have focused on scaling these methods to larger graphs. Beyond random sampling (Bevilacqua et al., 2021; Zhao et al., 2022), Qian et al. (2022) first proposed gradient-based techniques to *learn* how to subsample the bag of subgraphs. Bevilacqua et al. (2024) introduced *Policy-Learn* (PL), which iteratively predicts a distribution over nodes in the graph and samples subgraphs from the full-bag accordingly. A similar approach, called MAG-GNN has also proposed to sample subgraphs using Reinforcement Learning (RL) (Kong et al., 2024). Both of these approaches involve *discrete sampling* which can complicate the training process, often requiring $1,000 - 4,000$ training epochs. Another recent approach leverages the connection between Subgraph GNNs and graph products (Bar-Shalom et al., 2024a) to run message passing on the product of the original graph and a coarsened version thereof (Bar-Shalom et al., 2024b). The control over the computational complexity generally comes from the existence of cluster-like structures in the graph which, however, may not be aligned with the preset number of subgraphs. Additionally, the locality bias afforded by the coarsening may not generally be effective across tasks. In such cases, coarsening approaches that are not learnable and based, e.g., on spectral clustering could lead to suboptimal results.

**Node Centrality.** One common way to characterize nodes in a graph is by using the concept of *node centrality*. A node centrality measure defines a real-valued function on the nodes, $c : V \to \mathbb{R}$, which can be used to rank nodes within a network by their "importance". Different concepts of importance have led to a myriad of measures in the Network Science community. They range from the simple Degree Centrality (central nodes have the highest degrees) to path-based methods (Freeman, 1977) like the Betweenness Centrality (central nodes fall on the shortest paths between many node-pairs). An important family of centrality measures quantifies the importance of nodes based on *walk counts*. As noted in (Benzi & Klymko, 2014), most of these measures take the form of a power series, where numbers of walks for any lengths are aggregated with an appropriate discounting scheme. Prominent examples include the Katz Index (Katz, 1953) (KI) and the Subgraph Centrality (Estrada & Rodriguez-Velazquez, 2005) (SC):

$$c_i^{\text{KI}} = \sum_{k=0}^{\infty} \alpha^k \sum_j (A^k)_{ij} \qquad c_i^{\text{SC}} = \sum_{k=0}^{\infty} \frac{\beta^k}{k!} (A^k)_{ii} \tag{1}$$

and variants of the above, for appropriate choices of $0 < \alpha < \frac{1}{\lambda_1}, \beta > 0$[1] (Benzi & Klymko, 2013; 2014). By scoring nodes based on the cumulative number of walks that start from them, these centrality measures extend the Degree Centrality beyond purely local interactions, in a way that depends on the discounting scheme ($\alpha^k$ and $\frac{\beta^k}{k!}$ for KI and SC, respectively).

## 3 SUBSAMPLING SUBGRAPH NEURAL NETWORKS

### 3.1 PROBLEM SETTING

We focus on Subgraph GNNs with a node-marking selection policy (Papp & Wattenhofer, 2022; You et al., 2021). Given an $N$-node graph $G = (A, X)$, a node-marking Subgraph GNN processes a bag of subgraphs obtained from $G$, viz. $B_G = \{\!\{S_1, S_2, \ldots, S_N\}\!\}$. Here, $S_i = (A, X_i)$ and $X_i = X \oplus x_{v_i}$, where $\oplus$ denotes channel-wise concatenation and $x_{v_i}$ is a one-hot indicator vector for node $v_i$.

**Goal.** In order to reduce the computational complexity of a Subgraph GNN, we aim to reduce the size of the bag by *efficiently and effectively* sampling $k < N$ subgraphs.

---

[1] $\lambda_1$ refers to the first eigenvalue of $A$, $\beta$ is typically set to 1.

The sampling procedure must be *efficient* in that it should avoid computationally complex operations or the use of learnable components requiring more involved training protocols (Qian et al., 2022; Bevilacqua et al., 2024; Kong et al., 2024). Ideally, it should consist of a simple and lightweight preprocessing step prior to running a chosen Subgraph GNN. The sampling procedure must additionally be *effective*, which means it should closely approach the performance of a full-bag Subgraph GNN with as few subgraphs as possible. Put differently, it should prioritize marking those nodes that most quickly lead to performance improvements. As an example, to ground the discussion: randomly selecting subgraphs (Bevilacqua et al., 2021; Papp et al., 2021) is an efficient but not effective strategy; learning which subgraph to mark via RL (Kong et al., 2024) is a more effective approach, but it may not be efficient enough.

We start by presenting considerations on effectiveness which will naturally lead us to focus on walk-based centrality measures for our purposes. Upon them, we show we can build an effective strategy that is also efficiently executed as a simple preprocessing step.

## 3.2 Effective Node Marking

What makes a node a good marking candidate? We propose approaching this question by considering *node marking as a graph perturbation*. Marking a node changes the initial node features: this alteration in the input will ultimately be reflected in the output graph representation, and understanding how node marking impacts the output graph representation is instrumental in designing effective sampling strategies. Beyond binary graph separation, we claim that an effective marking should be able to (i) alter graph representations sufficiently; (ii) induce perturbations that correlate with graph targets. Ideally, when marking nodes jointly optimizes (i) and (ii), a Subgraph GNN with a small bag can then improve on a standard MPNN by sufficiently separating more graphs and in a way that assists the training objective.

**Node Marking, perturbations, and centrality measures**. To understand how marking a node alters graphs representations, we analyze the simplest case: marking a single node. In particular, we ask how much the representation of a generic graph $G$ can be changed by a single-node marking. As the MPNN, we consider an $L$-layer GIN (Xu et al., 2018; Chuang & Jegelka, 2022):

$$h_v^{(l)} = \phi^{(l)}\big(h_v^{(l-1)} + \epsilon \sum_{u \in N(v)} h_u^{(l-1)}\big) \qquad y_G = \phi^{(L+1)}\big(\sum_{v \in G} h_v^{(L)}\big) \qquad (2)$$

where $\phi^{(l)}$'s are update functions and $\phi^{(L+1)}$ is a prediction layer (all are parameterized as MLPs). By applying results from GNN stability studies in (Chuang & Jegelka, 2022) we put forward the following observation (see details in Appendix C):

**Observation 1.** *The distance between the MPNN representations of a graph $G$ and a graph $S_v$ generated by marking node $v$ in $G$, can be upper-bounded as:*

$$|y_G - y_{S_v}| \le \prod_{l=1}^{L+1} K_\phi^{(l)} \cdot \underbrace{\sum_{l=1}^{L+1} \lambda_l \cdot \sum_j (A^{l-1})_{v,j}}_{(\mathcal{A})} \qquad (3)$$

*where $K_\phi^l$ is the Lipschitz constant of MLP $\phi^{(l)}, l = 1 \dots L+1$, A is the adjacency matrix of graph $G$ and $\lambda_l \in \mathbb{R}^+$ is a layer-wise weighting scheme (dependent on $\epsilon$).*

*Effectively, the cumulative number $(\mathcal{A})$ of walks starting from node $v$ contributes to upper-bound the perturbation that marking $v$ induces on the original graph representation. Hence, marking nodes involved in a lower number of walks will have a more limited influence on altering a message passing-based graph representation.*

We note that the above observation uncovers an intriguing alignment with walk-based centrality measures: the "most important" nodes associate with the largest (discounted) cumulative numbers of walks (compare Equations 1 with term $(\mathcal{A})$ in Equation (3)). This leads us to direct our focus to walk-based centrality measures as promising candidates for selecting which node to mark. Considering we would like to sufficiently alter the graph representation beyond the one from a standard MPNN, Observation 1 indicates that, among all possible nodes, those associated with small cumulative

numbers of walks will be poor marking candidates. We propose to summarise this form of information via walk-based centrality measures. Our strategy will be to *rank nodes based on their centrality values, and mark the top-scoring ones*. In the following, we empirically verify the validity of this approach, while deferring readers to Appendix C for extensions and additional considerations on the above analysis.

**High-centrality marking induces the largest perturbations**. As a first experiment, we examine the extent to which node centralities recapitulate the amount of perturbation induced by marking their corresponding nodes. We consider the same setting discussed above: marking one node to transition from $G$ to $S_v$, where $S_v$ is obtained from $G$ by marking node $v$. For a centrality measure $c$, we consider three cases: $v$ attains the minimum of $c$ (i), the maximum of $c$ (ii), is randomly picked (iii). In each of these cases, we measured the distance $\|f(S_v) - f(G)\|$ on 100 graphs from two different real-world datasets from the popular TU suite: MUTAG and NCI1 (Morris et al., 2020a). Here, $f$ is an untrained 3-layer GIN (Xu et al., 2018). Figure 1 shows results for the walk-based Subgraph Centrality (Estrada & Rodriguez-Velazquez, 2005), where horizontal lines indicate the average representation distance, and (i), (ii), (iii) are color-coded, resp., in green, blue, red.

From the plots, it is clearly visible how marking nodes with the lowest centrality leads to the smallest change in graph representation. This result gives direct empirical validation to the upper-bound analysis and the consequential observation discussed above. In accordance with our claim, subgraphs associated with low centrality nodes can be interpreted as "redundant" w.r.t. the original input graph, or simply as poor marking candidates. Second, we note that marking nodes with the highest centrality induces the highest average perturbations, above random marking. This result is particularly relevant as it complements the above theoretical analysis: the walk-based upper-bound (Equation (3)) only suggests, but does not necessarily entail, that high-centrality marking associates with the largest perturbations. Figure 1 shows that this occurs in practice on these datasets, further motivating our proposed sampling strategy. Results for other centrality measures are found in Appendix D; they indicate that high-centrality marking leads, in all cases, to larger perturbations than random marking, with the highest values attained by walk-based centrality measures (see Section 2).

**High-centrality marking aligns with substructure counting**. We have shown how marking nodes with higher centrality can lead to larger graph perturbations. However, this may not be sufficient. As an example, consider, two (non-isomorphic but) 1-WL-equivalent graphs. Node marking can alter their message-passing-based representations, but not necessarily in a way to induce separation: ideally, if the two graphs are associated with different targets we aim to induce *dissimilar perturbations* so to assist the training goal. In effect, we want our sampling strategy to alter the graph representations in a way that is consistent with the target space. Motivated by the observation that the presence and number of structural "motifs" are often related to graph-level tasks (Kanatsoulis & Ribeiro, 2024), we empirically study how marking-induced perturbations correlate with counting small substructures, as a general, yet relevant, predictor for graph-level targets.

We randomly generate 100 Erdös-Renyi (ER) graphs, each with $N = 20$ nodes and wiring probability $p = 0.3$. Similarly as above, we experiment with various centrality measures, by marking a single node $v$ which attains either the maximum or minimum centrality value, or is randomly picked. Again, we record the perturbation $\|f(S_v) - f(G)\|$ given by the same architecture described above. On the same graphs, we count the number of various substructures, and evaluate the Pearson correlation between this value and the recorded perturbations. Results are reported in Table 1.

The top section of the table compares the correlations obtained by marking randomly or based on the Subgraph Centrality. The perturbations induced by high-centrality marking correlate significantly more with the considered substructures than those induced by low-centrality-based or random marking. The bottom section of the table presents results for other centralities not based on walks. We note how they all deliver better correlations w.r.t. random marking, but not as high as those attained by the SC.

**Discussion**. Overall our experiments indicate the following. First, *high-centrality sampling* appears to be a better approach than random sampling, especially when walk-based centrality measures are employed: on average, it selects marking candidates inducing the largest amount of perturbations over the original graph representations, and in a way that correlates with counts of relevant graph substructures. Second, the walk-based Subgraph Centrality stands out as a particularly promising candidate: it is efficient to precalculate this measure and sample subgraphs based on that, while, on average, it performed as the best one in the experiments discussed above.

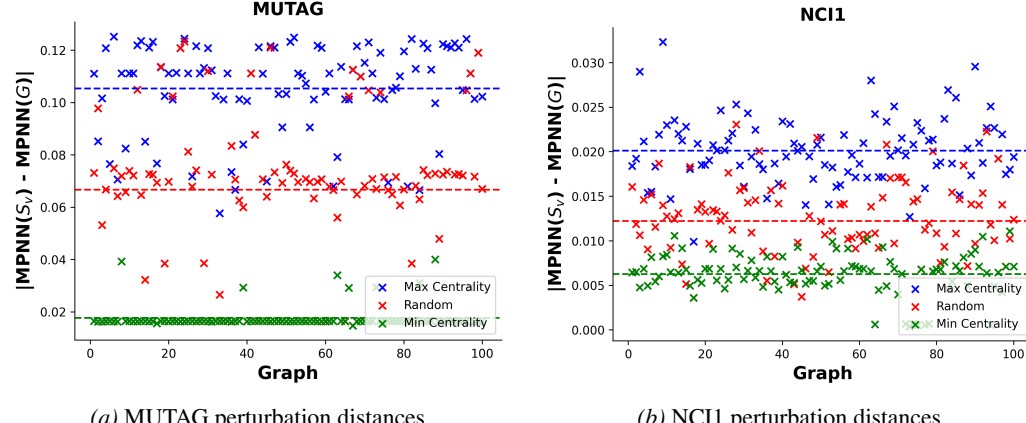

*(a)* MUTAG perturbation distances      *(b)* NCI1 perturbation distances

*Figure 1:* Plots showing the amount the graph representation using GIN is altered by adding an additional node-marked subgraph with (i) the highest centrality, (ii) the lowest centrality and (iii) a random marking. This is shown for both the MUTAG dataset (a) and for the NCI1 dataset (b).

*Table 1:* Pearson correlation between various substructure counts and amount of perturbation caused by different node-marking policies.

| Method | Tri ($\uparrow$) | 4-Cyc ($\uparrow$) | Tailed Tri ($\uparrow$) | Star ($\uparrow$) |
|---|---|---|---|---|
| Maximum Subgraph Centrality | **0.947** | **0.956** | **0.958** | **0.972** |
| Minimum Subgraph Centrality | 0.644 | 0.643 | 0.634 | 0.698 |
| Random | 0.712 | 0.708 | 0.723 | 0.723 |
| Maximum Degree Centrality | 0.937 | 0.947 | 0.948 | 0.962 |
| Maximum Closeness Centrality | 0.935 | 0.937 | 0.946 | 0.957 |
| Maximum Betweenness Centrality | 0.803 | 0.816 | 0.821 | 0.845 |

In the following, we will focus on this measure in particular. Our strategy will consist of marking only the top-ranking $k$ nodes according to SC, for a small, fixed $k$. As we will show, this simple approach already delivers strong empirical performance. Note that more sophisticated sampling schemes could be designed based on extensions of the above perturbation analysis. These could consider more complex Subgraph GNN architectures (Frasca et al., 2022; Zhang et al., 2023a) or study the effect of multiple node markings, for which a deeper inquiry could take into account pair-wise scores beyond node-wise centrality measures[2]. We defer these efforts to future work.

## 4   COMBINING SUBGRAPH GNNS WITH STRUCTURAL ENCODINGS

### 4.1   OUR APPROACH

**Subgraph Centrality as a SE**. In Section 3, we introduced the use of walk-based centrality measures, particularly Subgraph Centrality, as an efficient and effective method for subgraph sampling. These centrality measures can be expressed as power series expansions of the adjacency matrix (Benzi & Klymko, 2014) (see Equation (1)). We notice that addenda terms in the series already provide precious discriminative structural information, which could be desirable to employ for feature augmentations (Rampášek et al., 2022; Bouritsas et al., 2022; Dwivedi et al., 2021). Precisely, in the case of our chosen Subgraph Centrality, for the default $\beta = 1$, the $k$-th term $\frac{(A^k)_{vv}}{k!}$ is (the discounted number of) $k$-length closed-walks originating from $v$. As notable examples, these values are proportional to the degree of $v$ and the number of incident triangles for $k = 2, 3$. These considerations suggest retaining the intermediate values that contribute to the SC of each node, and employ them à la Structural Encodings beyond sampling purposes.

---

[2]For example, when selecting multiple nodes to mark, one could also account for the distance between them.

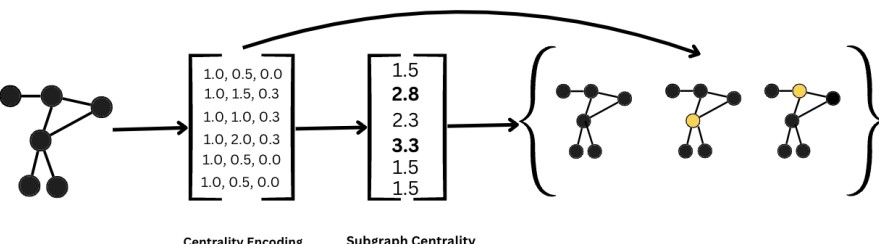

*Figure 2:* An overview of the pipeline for HyMN. We first calculate the centrality encoding for each node in the graph. We sum each row of this structural encoding to approximate the Subgraph Centrality of each node. We then sample $T$ node-marked subgraphs with the highest Subgraph Centrality on the marked node and concatenate the centrality encoding with the initial node features.

We naturally define the Centrality-based Structural Encodings (CSE) of order $K$ for node $v$ as[3]:

$$C_v^{\mathsf{CSE}} = \left[ 1, \frac{(A)_{vv}^1}{1!}, \frac{(A)_{vv}^2}{2!}, \frac{(A)_{vv}^3}{3!}, \dots, \frac{(A)_{vv}^K}{K!} \right] \in \mathbb{R}^{K+1}. \tag{4}$$

As $K \to \infty$, the sum of these terms clearly coincides with the SC of the node for the default $\beta = 1$. We refer readers to Appendix D.2, for considerations on how CSEs compare with RWSEs.

**Hybrid Marking Networks**. Our overall method consists in *jointly* (1) augmenting node features with $K$-order CSEs; (2) subsampling the $T$ node-marked subgraphs for the nodes attaining the highest centrality values; (3) processing the obtained bag of subgraphs with a Subgraph GNN of choice. In view of (1) and (2), we dub our approach HyMN, as in *Hybrid Marking Network*. These steps are depicted in Figure 2 and described in Algorithm 1. We note the following. First, for a large enough $K$, centrality values can be approximated by directly summing over the computed $K$-order CSEs. Second, node marking does not require any alteration of the original graph topology, making it unnecessary to store the subgraph connectivity. We thus opt not to materialize the bag of subgraphs: we only record marking information in the feature tensor and implement a custom message-function that processes it in an equivariant way. From an engineering standpoint, this allows for further memory-complexity enhancement w.r.t. generic Subgraph GNN approaches.

---

**Algorithm 1** Hybrid Marking Network

---

**Require:** Graph $G = (A, X)$, Subgraph GNN $f$, Max walk-length $K$, Number of marks $T$
1: $C_{v,k}^{\mathsf{CSE}} \leftarrow 0 \quad \forall v \in G, k \in [K]$      ▷ SE init.
2: $M_{v,t} \leftarrow 0 \quad \forall v \in G, t \in [T]$     ▷ Mark init.
3: $B \leftarrow I$
4: **for** $k \in [K]$ **do**
5:     **for** $v \in G$ **do**
6:         $C_{v,k}^{\mathsf{CSE}} \leftarrow {}^{B_{vv}}/_{k!}$    ▷ Compute SEs
7:     **end for**
8:     $B \leftarrow B \cdot A$
9: **end for**
10: $\tilde{C}^{SC} \leftarrow \sum_k C_{:,k}^{\mathsf{CSE}}$       ▷ Estimate SC
11: $\mathcal{M} \leftarrow \text{select-top}(\tilde{C}^{SC}, T)$   ▷ Select nodes
12: **for** $t \in [T]$ **do**
13:     $M_{\mathcal{M}[t],t} \leftarrow 1$         ▷ Mark nodes
14: **end for**
15: $y_G \leftarrow f(A, X \oplus C^{\mathsf{CSE}}, M)$   ▷ Forward-pass
16: **return** $y_G$

---

### 4.2 Expressivity of Subgraph GNNs with Centrality Encodings

HyMN effectively marries two distinct Graph Learning approaches: the use of node-marked subgraphs and of SEs. At this point, it is natural to ask whether this combination of techniques is justified from an expressiveness perspective. Put differently, we ask whether enhancing message passing with CSEs already subsumes our high-centrality marking strategy or, vice-versa, whether our marking approach could recover CSEs.

We answer these questions with graph separation arguments (Xu et al., 2018; Morris et al., 2019) and highlight how, in fact, the two approaches are not generally comparable. We demonstrate that subsampling Subgraph GNNs with high-centrality marking *does not* subsume CSE-enhanced MPNNs, while, at the same time, the discriminative power of the former approach *is not* fully captured by the

---

[3]We defined CSEs directly as the addenda in the power-series in Equation (1), but we note that the first two terms are not discriminative and could be dropped.

latter. Since neither technique alone fully subsumes the other, our analysis emphasizes the advantages of combining them in HyMN for improved expressiveness. Proofs and additional details are reported in Appendix B.

**MPNNs with centrality encoding do not subsume subsampled Subgraph GNNs**. Below, we show that node-marked subgraphs can separate graphs indistinguishable by CSE-enhanced MPNNs, i.e., MPNNs running on graphs whose features are augmented with our centrality-based encodings.

**Theorem 1.** *There exists a pair of graphs $G$ and $G'$ such that for any CSE-enhanced MPNN model $M_{CSE}$ we have $M_{CSE}(G) = M_{CSE}(G')$, but there exists a DS-Subgraph GNN model (without CSEs) $M_{sub.}$ which uses a top-1 Subgraph Centrality policy such that $M_{sub.}(G) \neq M_{sub.}(G')$.*

This result is proved, in particular, by considering two 1-WL equivalent graphs which have identical values for CSEs. This makes them indistinguishable by a CSE-enhanced MPNN, contrary to (sampled) DS-Subgraph GNNs (Bevilacqua et al., 2021), the simplest Subgraph GNN variants which process subgraphs independently. This underscores the advantage of incorporating a node-marking Subgraph GNN alongside structural encoding techniques.

**Subsampled Subgraph GNNs do not subsume MPNNs with centrality encoding**. Processing only a fixed number of subgraphs selected by our high-centrality strategy may limit discriminative power. In particular, the following shows that this approach does not subsume CSE-enhanced MPNNs:

**Theorem 2.** *There exists a pair of graphs $G$ and $G'$ such that for any Subgraph GNN model $M_{sub.}$ which uses a top-1 Subgraph Centrality policy we have $M_{sub.}(G) = M_{sub.}(G')$, but there exists an MPNN + centrality encoding model $M_{CSE}$ such that $M_{CSE}(G) \neq M_{CSE}(G')$.*

This result exposes a limitation of subsampled Subgraph GNNs in distinguishing between two non-isomorphic graphs with differing closed walks, features which are, instead, captured by CSEs. Notably, as discussed in Proposition 2 (Appendix B), a *full-bag* approach is capable of capturing CSEs. This observation suggests that while CSEs do not universally enhance the expressiveness of any Subgraph GNN, they can be beneficial when subsampling a limited number of subgraphs.

Taken together, Theorems 1 and 2 indicate that leveraging SC both as a structural encoding *and* as a means for subgraph sampling is advantageous from an expressiveness perspective, justifying the integration of the two techniques in HyMN.

## 5 EXPERIMENTS

Our experiments aim to validate arguments in the previous sections and to empirically answer the following questions:

(**Q1**) *Can Subgraph Centrality be used to effectively subsample subgraphs for Subgraph GNNs?*
(**Q2**) *Can HyMN efficiently scale to graphs out of reach for Subgraph GNNs?*
(**Q3**) *How does HyMN perform on real-world datasets w.r.t. strong GNN baselines?*
(**Q4**) *What is the impact of incorporating CSEs?*

**Synthetic experiment for counting substructures**. The ability of a model to count local substructures is an acknowledged way of evaluating its expressive power (Bouritsas et al., 2022; Arvind et al., 2020; Tahmasebi et al., 2020). In order to answer (**Q1**) and test the efficacy of subgraph sampling with node centrality, we explored the ability of a Subgraph GNN to count different small substructures as we increase the number of subgraphs in our bag. We closely followed the experimental procedure of (Chen et al., 2020), but modified the data generation process to render the task more challenging and informative[4]. We compared the performance of sampling subgraphs based on different approaches: random sampling and sampling based on the highest values of different centrality measures. No CSEs are employed in this setting. Section 5 reports results for triangle and 4-cycle counting. These results demonstrate the significant improvement afforded by high-centrality sampling over random sampling with fewer subgraphs (**Q1**). Additionally, we show that SC-based sampling generally outperformed other centrality measures, demonstrating the benefits of focusing on walk-based centralities, in alignment with our analysis in Section 3. Appendix D reports results for additional substructures,

---

[4]In particular, we considered larger graphs with a similar number of nodes to correct an undesired correlation between the graph size and the task targets observed in the original data (see Appendix E.2).

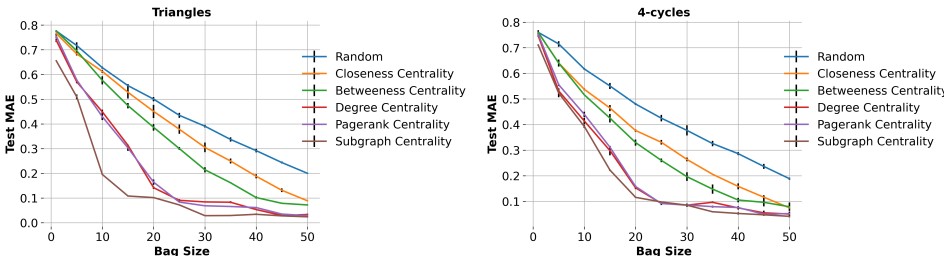

*Figure 3:* Comparing Random and Centrality-based sampling on counting different substructures in the graph. For reference, the average graph size is 59.33, corresponding to the average bag-size of a full-bag Subgraph GNN.

*Table 2:* Results on OGB datasets. The first and second best results for each task are color-coded.

| Method | MOLHIV (ROC-AUC ↑) | MOLBACE (ROC-AUC ↑) | MOLTOX21 (ROC-AUC ↑) |
|---|---|---|---|
| GCN Kipf & Welling (2017) | 76.06 ±0.97 | 79.15 ±1.44 | 75.29 ±0.69 |
| GIN Xu et al. (2018) | 75.58 ±1.40 | 72.97 ±4.00 | 74.91 ±0.51 |
| FULL Bevilacqua et al. (2024) | 76.54 ±1.37 | 78.41 ±1.94 | 76.25 ±1.12 |
| OSAN Qian et al. (2022) | - | 76.30 ±3.00 | - |
| MAG-GNN Kong et al. (2024) | 77.12 ±1.13 | - | - |
| RANDOM (T = 2) Bevilacqua et al. (2024) | 77.55 ±1.24 | 75.36 ±4.28 | 76.65 ±0.89 |
| POLICY-LEARN (T = 2) Bevilacqua et al. (2024) | 79.13 ±0.60 | 78.40 ±2.85 | 77.47 ±0.82 |
| **HyMN (GIN, T=2) w/out CSE** | 79.77 ±0.70 | 78.22 ±4.02 | **77.68** ±0.71 |
| **HyMN (GIN, T=2)** | **81.01** ±1.17 | **81.16** ±1.21 | 77.30 ±0.35 |
| RANDOM (T=5) Bevilacqua et al. (2024) | 77.30 ±2.56 | 78.14 ±2.36 | 76.62 ±0.89 |
| POLICY-LEARN (T=5) Bevilacqua et al. (2024) | 78.49 ±1.01 | 78.39 ±2.28 | 77.36 ±0.60 |
| **HyMN (GIN, T=5) w/out CSE** | 79.62 ±1.14 | 78.57 ±1.31 | **77.82** ±0.59 |
| **HyMN (GIN, T=5)** | **80.17** ±1.40 | **79.94** ±0.48 | 76.99 ±0.45 |

real-world experiments, and other centrality measures where the effectiveness of SC-based sampling is further demonstrated.

**OGB**. We tested HyMN on several datasets from the OGB benchmark (Hu et al., 2020b). To further examine (**Q1**), Table 2 shows the performance of our approach in relation to MPNNs, a full-bag Subgraph GNN and other subgraph sampling policies with the same number of subgraphs. Notably, even without using the centrality-based encoding (HyMN w/out CSE), our method matches the performance of a learnable sampling policy (POLICY-LEARN (Bevilacqua et al., 2024)) and consistently outperforms MPNNs and random sampling policies. Additionally, we observe that augmenting node features with CSEs can significantly increase performance on MOLHIV and MOLBACE, outperforming a full-bag Subgraph GNN (**Q4**). These results suggest that centrality sampling is effective and that additionally incorporating centrality information can lead to performance improvements on real-world datasets, aligning with our findings in Section 3. Additional results for MOLHIV are reported in Table 4, where we show HyMN can outperform strong GNN baselines (**Q3**).

**Peptides**. In order to evaluate the ability of HyMN to scale to larger graphs (**Q2**), we experimented on the Peptides datasets from the LRGB benchmark (Dwivedi et al., 2022). The average number of nodes in these graphs is 150.94, so it is difficult for a full-bag Subgraph GNN to process. Additionally, using the centrality encoding to sample just one or two additional node-marked subgraphs can improve performance on both datasets. We also outperform GPS (Rampášek et al., 2022) (a Graph Transformer) and match the performance of Graph-MLP-Mixer (He et al., 2023) (**Q3**). Our timing experiments on Peptides-Func demonstrate that we are significantly more efficient than both of these approaches, only increasing the training time per epoch over a GCN by 42% and the inference time by 25% using 'HyMN (GCN, T=1)' (**Q2**).

**Zinc**. We experimented with the ZINC-12K molecular dataset (Sterling & Irwin, 2015; Gómez-Bombarelli et al., 2018), where we maintain a 500k parameter budget, in line with previous works. We compared the test MAE using HyMN to other sampling approaches (Qian et al., 2022; Bevilacqua et al., 2024), a full-bag Subgraph GNN, a Graph Transformer (GPS) (Rampášek et al., 2022) and two expressive GNN baselines (GSN, CIN) (Bouritsas et al., 2022; Bodnar et al., 2021a). As can be seen

*Table 3:* Results on Peptides datasets with timing comparisons on Peptides-Func using a NVIDIA GeForce RTX 3080 10GB. Test AP is quoted for Peptides-Func and Test MAE for Peptides-Struct.

| Method | Precompute (s) | Train (s/epoch) | Test (s) | Peptides-Func ($\uparrow$) | Peptides-Struct ($\downarrow$) |
|---|---|---|---|---|---|
| GIN | 0.00 $\pm$0.00 | 2.65 $\pm$0.01 | 0.238 $\pm$0.004 | 0.6555 $\pm$0.0088 | 0.2497 $\pm$0.0012 |
| GIN + CSE | 20.12 $\pm$0.39 | 2.78 $\pm$0.01 | 0.253 $\pm$0.004 | 0.6619 $\pm$0.0077 | 0.2479 $\pm$0.0011 |
| **HyMN (GIN, T=1)** | 23.71 $\pm$0.34 | 4.93 $\pm$0.03 | 0.420 $\pm$0.002 | 0.6857 $\pm$0.0055 | 0.2464 $\pm$0.0013 |
| **HyMN (GIN, T=2)** | 23.75 $\pm$0.32 | 6.60 $\pm$0.03 | 0.561 $\pm$0.001 | 0.6863 $\pm$0.0050 | **0.2457** $\pm$0.0012 |
| GCN | 0.00 $\pm$0.00 | 2.07 $\pm$0.04 | 0.234 $\pm$0.006 | 0.6739 $\pm$0.0024 | 0.2505 $\pm$0.0023 |
| GCN + CSE | 20.19 $\pm$0.36 | 2.16 $\pm$0.04 | 0.254 $\pm$0.005 | 0.6812 $\pm$0.0037 | 0.2499 $\pm$0.0010 |
| **HyMN (GCN, T=1)** | 23.88 $\pm$0.30 | 2.94 $\pm$0.01 | 0.292 $\pm$0.002 | **0.6912** $\pm$0.0170 | 0.2481 $\pm$0.0013 |
| **HyMN (GCN, T=2)** | 23.97 $\pm$0.30 | 3.83 $\pm$0.01 | 0.368 $\pm$0.002 | **0.6948** $\pm$0.0052 | 0.2477 $\pm$0.0010 |
| GPS (Rampášek et al. (2022)) | 20.87 $\pm$0.43 | 8.39 $\pm$0.05 | 0.611 $\pm$0.005 | 0.6535 $\pm$0.0041 | 0.2500 $\pm$0.0005 |
| Graph-ViT (He et al. (2023)) | 29.12 $\pm$0.61 | 6.78 $\pm$0.01 | 0.709 $\pm$0.009 | **0.6942** $\pm$0.0075 | **0.2449** $\pm$0.0016 |
| G-MLP-Mixer (He et al. (2023)) | 29.52 $\pm$0.69 | 6.87 $\pm$0.03 | 0.684 $\pm$0.003 | **0.6921** $\pm$0.0054 | 0.2475 $\pm$0.0015 |

from Table 4, our hybrid method can outperform a full-bag Subgraph GNN as well as previously proposed subsampling based approaches. Additionally, we perform competitively with CIN, which takes into account higher-order interactions and explicitly models ring-like structures. We highlight that additionally using subgraphs can outperform purely using the centrality-based encodings (**Q4**).

**Summary**. In reference to the questions enlisted above, we conclude the following. (**A1**) The results from substructure counting and on the OGB benchmarks suggest a positive answer to **Q1**: SC-based sampling significantly outperformed random sampling on both, and matched the performance of the learnable POLICY-LEARN. (**A2**) HyMN was easily applied to the larger Peptides datasets, with strong empirical performance and favorable inference and training run-times. This indicates our approach broadens the applicability of Subgraph GNNs to larger graphs. (**A3**) Beyond Peptides, HyMN also attained remarkable performance on OGB benchmarks, and performed competitively on ZINC. This suggests a generally positive answer to **Q3**. (**A4**) We observe that CSEs can enhance the performance of standard MPNNs (see Table 3) and subsampled Subgraph GNNs (see, e.g., MOLHIV and MOLBACE in Table 2). We note, however, they were not beneficial on MOLTOX21 (Table 2).

## 6 CONCLUSION

We introduced a novel framework, termed HyMN, which combines a subgraph sampling strategy and structural encodings both derived from walk-based centrality measures and, in particular, the Subgraph Centrality by (Estrada & Rodriguez-Velazquez, 2005). We showed that this centrality is a good measure of subgraph importance: for a very limited number of subgraphs it enables competitive performance and outperforms random and learnable selection strategies, approaching full-bag methods. The additional inclusion of centrality-based SEs is also proved to be beneficial both theoretically and in practice, allowing to enhance downstream generalization performance on several real-world benchmarks. Importantly, the strong performance of our method is achieved with a thin computational overhead, making it applicable to a wider spectrum of downstream tasks.

*Table 4:* Test results on the ZINC (Sterling & Irwin, 2015) and MOLHIV (Hu et al., 2020b) datasets. The first and second best results for each task are color-coded.

| Method | ZINC ($\downarrow$) | MOLHIV ($\uparrow$) |
|---|---|---|
| GCN | 0.321 $\pm$0.009 | 76.06 $\pm$0.97 |
| GIN | 0.163 $\pm$0.004 | 75.58 $\pm$1.40 |
| GSN | 0.101 $\pm$0.010 | 80.39 $\pm$0.90 |
| CIN | 0.079 $\pm$0.006 | 80.94 $\pm$0.57 |
| GPS | 0.070 $\pm$0.004 | 78.80 $\pm$1.01 |
| GINE-MLP-Mixer | 0.073 $\pm$0.001 | 79.97 $\pm$1.02 |
| GINE-ViT | 0.085 $\pm$0.004 | 77.92 $\pm$1.42 |
| FULL | 0.087 $\pm$0.003 | 76.54 $\pm$1.37 |
| OSAN | 0.177 $\pm$0.016 | - |
| POLICY-LEARN (T = 2) | 0.120 $\pm$0.003 | 79.13 $\pm$0.60 |
| RANDOM (T = 2) | 0.136 $\pm$0.005 | 77.55 $\pm$1.24 |
| GIN+CSE | 0.092 $\pm$0.002 | 77.44 $\pm$1.87 |
| **HyMN (GIN, T=1)** | 0.080 $\pm$0.003 | 80.36 $\pm$1.23 |
| **HyMN (GIN, T=2)** | 0.083 $\pm$0.002 | 81.01 $\pm$1.17 |

**Limitations and Future Work**. Our sampling procedure does not take into account already sampled subgraphs unlike methods such as the ones in (Zhao et al., 2022; Bevilacqua et al., 2024). Future work could focus on more general perturbation analyses to give an indication on multi-node marking for higher-order selection policies (Qian et al., 2022) or to quantify the impact of adding subgraphs to a partially populated bag. More sophisticated selection strategies could combine different walk-based centrality measures or consider pairwise structural features.

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

## A  OUTLINE OF THE APPENDIX

We first report proofs pertaining to Section 4 in Appendix B. We then detail the perturbation analysis of Section 3 in Appendix C. Furthermore, we conduct further experiments comparing Subgraph Centrality with different centrality measures and to RWSE in Appendix D. We then comprehensively describe our experiments, model and parameters in Appendix E.

## B  CLAIMS AND PROOFS

**Proposition 1.** *Let $G = (A, X)$ be a connected graph (with $n$ nodes) with initial node features $X_i = \mathbf{p}_i^{CSE}$. There exists an $L = k + 1$ layered Message Passing Neural Network (MPNN) that processes the graph $G$ and can compute the following structural encodings capturing closed-walk probabilities for walks of size up to $k$ for any node $i$, defined as:*

$$\boldsymbol{c}_i = \left[1, \frac{(A)_{ii}^1}{\sum_{j=1}^n (A)_{ij}^1}, \frac{(A)_{ii}^2}{\sum_{j=1}^n (A)_{ij}^2}, \dots, \frac{(A)_{ii}^k}{\sum_{j=1}^n (A)_{ij}^k}\right],$$

*up to arbitrary precision.*

*Proof.* We define a Message Passing Neural Network (MPNN) as a composition of layers of the form:

$$\tilde{X}^{(l)} = AX^{(l-1)}\mathbf{W}^{(l)_1} + X^{(l-1)}\mathbf{W}^{(l)_0}, \tag{5}$$

$$X^{(l)} = f^{(l)}(\tilde{X}^{(l)}) \tag{6}$$

where $\mathbf{W}^{(l)_1}$ and $\mathbf{W}^{(l)_0}$ are learned weight matrices, and $f^{(l)}$ is an MLP for the $l$-th layer. We refer to the layers in Eqs. (5) and (6) as the MPNN layer and MLP layer, respectively, where the MLP layer is assumed to be a single hidden layer with interleaved ReLU activations.

We recall that the initial node feature matrix, $X^{(0)} \in \mathbb{R}^{n \times (k+1)}$, is given as follows:

$$X^{(0)} = \begin{pmatrix} - & \mathbf{p}_1^T & - \\ - & \mathbf{p}_2^T & - \\ & \vdots & \\ - & \mathbf{p}_n^T & - \end{pmatrix}$$

where,

$$\mathbf{p}_i = \left[1, \frac{(A)_{ii}^1}{1!}, \frac{(A)_{ii}^2}{2!}, \frac{(A)_{ii}^3}{3!}, \dots, \frac{(A)_{ii}^k}{k!}\right] \in \mathbb{R}^{k+1}.$$

**Step 1: Recovering the unnormalized CSE.** To uncover the unnormalized CSE, we set $\mathbf{W}^{(1)_1} = 0$ and $\mathbf{W}^{(1)_0} \in \mathbb{R}^{(k+1)\times((k+1)+k)}$ as follows:

$$\mathbf{W}^{(1)_0} = \begin{pmatrix} 1 & 0 & 0 & \cdots & 0 & 0 & \cdots & 0 \\ 0 & 1! & 0 & \cdots & 0 & 0 & \cdots & 0 \\ 0 & 0 & 2! & \cdots & 0 & 0 & \cdots & 0 \\ \vdots & \vdots & \vdots & \ddots & \vdots & \vdots & \vdots \\ 0 & 0 & 0 & \cdots & k! & 0 & \cdots & 0 \end{pmatrix}$$

Thus, the MPNN layer gives:

$$\tilde{X}^{(1)} \triangleq \mathrm{MPNN}^{(1)}(A, X) = \begin{pmatrix} - & \mathbf{p}_1^{\mathrm{unnormalized}^T} & - \\ - & \mathbf{p}_2^{\mathrm{unnormalized}^T} & - \\ & \vdots & \\ - & \mathbf{p}_n^{\mathrm{unnormalized}^T} & - \end{pmatrix}$$

where,

$$\mathbf{p}_i^{\mathrm{unnormalized}^T} = \left[1, (A)_{ii}^1, (A)_{ii}^2, (A)_{ii}^3, \ldots, (A)_{ii}^k, 0, \ldots, 0\right] \in \mathbb{R}^{(k+1)+k}.$$

For the MLP $f^0$, we use an identity weight matrix and a bias vector, which is all zeros except for the last $k$ values, which are ones:

$$\mathbf{b}^{(0)} = [0, 0, \ldots, 0, 1, 1, \ldots, 1] \in \mathbb{R}^{(k+1)+k}.$$

Thus, we obtain $X^{(1)}$, such that:

$$X_i^{(1)} = \left[1, (A)_{ii}^1, (A)_{ii}^2, (A)_{ii}^3, \ldots, (A)_{ii}^k, 1, \ldots, 1\right] \in \mathbb{R}^{(k+1)+k}.$$

**Step 2: Compute the matrix of closed-walk probabilities.** We compute the matrix of closed-walk probabilities in sequential steps. For each $j$-th step, we use the following weight matrices:

$$\mathbf{W}^{(j+1)_0} = \left(\begin{array}{c|c|c} I_{k+1} & 0 & 0 \\ \hline 0 & I_{j-1} & 0 \\ \hline 0 & 0 & 0_{k-(j-1)} \end{array}\right)$$

$$\mathbf{W}^{(j+1)_1} = \left(\begin{array}{c|c|c} 0_{k+1} & 0 & 0 \\ \hline 0 & 0_{j-1} & 0 \\ \hline 0 & 0 & I_{k-(j-1)} \end{array}\right)$$

Where $I_n$ is an $n \times n$ identity matrix, and $0_n$ is an $n \times n$ zero matrix (if $n = 0$ the corresponding block is omitted). This means:

$$X\mathbf{W}^{(j+1)_0}$$

keeps only the first $k + j$ columns of $X$, and,

$$AX\mathbf{W}^{(j+1)_1}$$

multiplies by $A$ only the last $k - (j - 1)$ columns of $X$.

By doing this iteratively for $j = 1$ to $j = k$, and setting the interleaved MLPs to identity weight matrices, we obtain the node matrix $\tilde{X}^{(k+1)}$, such that:

$$\tilde{X}_i^{(k+1)} = \left[1, (A)_{ii}^1, (A)_{ii}^2, (A)_{ii}^3, \ldots, (A)_{ii}^k, \sum_{j=1}^n (A)_{ij}^1, \sum_{j=1}^n (A)_{ij}^2, \ldots, \sum_{j=1}^n (A)_{ij}^k\right] \in \mathbb{R}^{(k+1)+k}.$$

Let $F : \mathbb{R}^{2k+1} \to \mathbb{R}^{k+1}$ be the following continuous function:

$$F(\mathbf{x})_i = \begin{cases} 1, & \text{if } i = 0, \\ \frac{\mathbf{x}_i}{\mathbf{x}_{k+i}}, & \text{if } i \neq 0. \end{cases}$$

Since the graph is connected, the denominator is always non-zero, making the function continuous. Additionally, since we are considering a finite graph with $n$ nodes, the input to the function $F$ lies within a compact set.

Using the universal approximation theorem Hornik (1991); Cybenko (1989), $F$ can be approximated to an arbitrary precision using an MLP. Thus, we use the MLP of layer $k + 1$ to realize $F$, obtaining the following node matrix,

$$X_i^{(k+1)} = \left[1, \frac{(A)_{ii}^1}{\sum_{j=1}^n (A)_{ij}^1}, \frac{(A)_{ii}^2}{\sum_{j=1}^n (A)_{ij}^2}, \cdots, \frac{(A)_{ii}^k}{\sum_{j=1}^n (A)_{ij}^k}\right] \in \mathbb{R}^{k+1}.$$

This completes the proof. $\qquad\qquad\square$

**Theorem 3** (Theorem 1 in Section 4). *There exists a pair of graphs $G$ and $G'$ such that for any CSE-enhanced MPNN model $M_{CSE}$ we have $M_{CSE}(G) = M_{CSE}(G')$, but there exists a DS-Subgraph GNN model (without CSEs) $M_{sub.}$ which uses a top-$k$ Subgraph Centrality policy such that $M_{sub.}(G) \neq M_{sub.}(G')$.*

*Proof.* Using the notation of Read & Wilson (1998), let $G, G'$ be the quartic vertex transitive graphs Qt15 and Qt19 respectively (Here vetrex transitive means that for each pair of nodes there exists a graph automorphism that maps one node to the other, and quartic refers to 4-regular). As these graphs are 4-regular and of the same size, they are 1-WL indistinguishable. In addition, as they are vertex transitive, for each pair of indices $i, j$ we have:

$$A_{i,i}^k = A_{j,j}^k = \frac{\text{trace}(A^k)}{12}. \tag{7}$$

$$A_{i,i}'^k = A_{j,j}'^k = \frac{\text{trace}(A'^k)}{12}. \tag{8}$$

Here the last equalities hold because both graphs have 12 vertices. Thus, to show that $G$ and $G'$ are indistinguishable by MPNN + centrality encoding it is enough to show that $\text{trace}(A^k) = \text{trace}(A'^k)$. As $G$ and $G'$ were shown in Brouwer & Spence (2009) to be co-spectral (i.e. their laplacian has the same eigenvalues) and 4-regular, matrices $A$ and $A'$ have the same eigenvalues. Thus we have:

$$\text{trace}(A)^k = \sum_{i=1}^{1} 2\lambda_i^k = \text{trace}(A')^k. \tag{9}$$

Here $\lambda_i$ is the $i$-th eigenvalue of both $A$ and $A'$. Thus the central encoding of all nodes in either graph is equal, and they are indistinguishable by any any MPNN + CE model. On the other hand, we observe that the degree histogram in the 1-hop neighborhood of any node differs between the two graphs, Qt15 and Qt19. Since an MPNN over a graph with a marked node can compute the degree distribution of the node's 1-hop neighborhood, $M_{\text{subgraph}}$ can distinguish between the two graphs. This concludes the proof.

$\qquad\qquad\square$

**Theorem 4** (Theorem 2 in Section 4). *There exists a pair of graphs $G$ and $G'$ such that for any Subgraph GNN model $M_{sub.}$ which uses a top-$k$ Subgraph Centrality policy we have $M_{sub.}(G) = M_{sub.}(G')$, but there exists an MPNN + centrality encoding model $M_{CSE}$ such that $M_{CSE}(G) \neq M_{CSE}(G')$.*

*Proof.* We begin by examining the scenario where $k = 1$, meaning that our policy randomly selects subgraphs corresponding to the node with the highest centrality measure. Consider the graph $G$, which is formed by attaching a global node to every vertex of a cyclic graph of length 6. Next, define $G'$ as the graph obtained by attaching a global node to each vertex of two disconnected cyclic graphs, each of length 3 (The global node is also attached to itself through a self loop). These graphs are displayed in Fig. 5. It can be easily seen that $G$ and $G'$ are WL indistinguishable (e.g. by induction). We first prove that both in $G$ and $G'$ the global node has the highest centrality measure. This implies that for both graphs, the resulting bag of subgraphs is of size one and is thus equivalent to standard message passing on the graphs (here we can ignore marking as the global nodes have a unique degree and so they can be uniquely identified by standard message passing). This implies that the two graphs are indistinguishable by any Subgraph GNN model $M_{\text{subgraph}}$ which uses a top-1 node centrality policy. We then show that the multiset of values of the centrality encoding of each graph is different, showing that it can be distinguished by an MPNN + centrality encoding model. To show that in both graphs the global node has the higher centrality, we first prove the following lemma:

**Lemma 1.** *Let $A$ denote the adjacency graph of one of the above graphs, $v$ denote the global node and $u_1, u_2$ denote a pair of nodes of the graph such that $u_1 \neq v$ . For each $k \in \mathbb{N}$ we have:*

$$A_{v,u_2}^k \geq A_{u_1,u_2}^k$$
$$A_{v,u_2}^k > 0 \tag{10}$$
$$A_{v,v}^k > A_{u_1,u_1}^k.$$

*Proof of lemma.* We use induction on $k$. As $v$ is connected to all nodes including itself , for $k = 1$, $A_{v,u_1} = 1$ . Since, disregarding the global nodes, $G, G'$ are simple graphs , we have $A_{u_1,u_1} = 0, A_{u_2,u_1} \leq 1$, thus the base case holds. Assuming the induction hypothesis holds for some $k$, we first notice that

$$A_{u_2,v}^{k+1} = A_{u_2,:}^k \cdot A_{:,v} \geq A_{u_2,:}^k \cdot A_{:,u_1} = A_{u_2,u_1}^{k+1}. \tag{11}$$

Here, $A_{u,:}^k, A_{:,u}^k$ represents the column/row vectors induced by node $u$ respectively and $\cdot$ denotes inner product. The inequality above follows from our induction hypothesis and the fact that all entries of the matrix $A^k$ are non-negative. Next, we notice that

$$A_{u_2,v}^{k+1} = A_{u_2,:} \cdot A_{:,v}^k \geq A_{u_2,v} \cdot A_{v,v}^k > 0. \tag{12}$$

In addition, we notice that $A_{v,u_1}^k \cdot A_{u_1,v} > 0 = A_{u_1,u_1}^k \cdot A_{u_1,u_1}$, where the last equallity holds because $A_{u_1,u_1} = 0$. Thus, we get:

$$A_{v,v}^{k+1} = A_{v,u_1}^k \cdot A_{u_1,v} + \sum_{u \neq u_1} A_{v,u}^k \cdot A_{u,v} > A_{u_1,u_1}^k \cdot A_{u_1,u_1} + \sum_{u \neq u_1} A_{u_1,u}^k \cdot A_{u,u_1} = A_{u_1,u_1}^{k+1}. \tag{13}$$

This completes the induction step.

$\square$

As explained before, the last lemma shows that top-1 centrality node marking policy always produces a bag with a single graph where the global node is marked. As the global node can be uniquely distinguished by its degree, this shows that a Subgraph GNN with this policy is equivalent to standard message passing and is thus unable to distinguish $G$ and $G'$. Finally, computing the centrality encoding of order 3 we see the multiset of features of the two graphs are different and so message passing + CE is able to seperate $G$ and $G'$.

We now address the general case of a top-$k$ centrality node-marking policy. Let $G_k, G_k'$ denote the graphs consisting of $k$ disjoint copies of $G$ and $G'$, respectively. In each disjoint copy, the global node is replicated independently and maintains a higher centrality than all other nodes within that copy. Thus, in both graphs, the k nodes with the highest centrality are the $k$ copies of the global node.

The bag of graphs generated from $G_k$ and $G_k'$ using the top-k centrality node-marking policy are then composed of $k$ copies of $G_k$ and $G_k'$ respectively, where a single copy of the global node is marked. Notice that in each one of these bags, all graphs are isomorphic to each other, thus it is enough to show that $G_k$ with a single marked global node copy is 1-WL indistinguishable from $G_k'$ with a single marked global node copy. To see this holds, notice that as we have seen above, the connected component of $G_k$ containing the marked node is 1-WL indistinguishable from the copy of the connected component of $G_k'$ containing the marked node, and the $k - 1$ unmarked connected components of $G_k$ are 1-WL indistinguishable from the unmarked connected components of $G_k'$. Thus any subgraph GNN which uses a top-k centrality node marking policy is unable to distinguish $G_k$ and $G_k'$. Finally, the centrality encoding values of each node $u_{\text{copy}}$ in $G_k$ is equal to the centrality encoding value of the node $u$ in $G$ which corresponds to $u_{\text{copy}}$. As we have seen before the set of centrality encoding values of $G$ and $G'$ are different, the set of centrality encoding values of $G_k$ and $G_k'$ are also different, and so message passing + CE is able to separate $G_k$ and $G_k'$.

$\square$

**Proposition 2.** *Let $G = (A, X)$ be a finite graph, and let $\mathcal{B}_G$ be the bag generated from original graph $G$. Let DSS-GNN be the subgraph-based GNN that processes the bag $\mathcal{B}_G$. There exists a set of weights for DSS-GNN, such that DSS-GNN$(\mathcal{B}_G) = $ MPNN$(G)$ for any MPNN processing $G$ with centrality-based structural encodings as initial node features.*

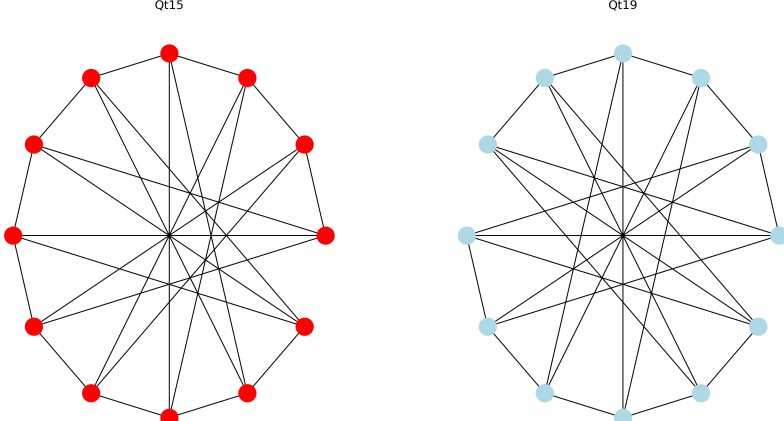

*Figure 4:* Two quartic vertex transitive graphs which cannot be distinguished with MPNN + CSE but can be distinguished with a Subgraph GNN with a top-1 Subgraph Centrality policy.

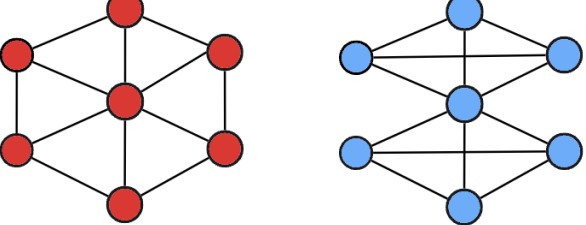

*Figure 5:* Two graphs which cannot be distinguished by a Subgraph GNN with a top-1 Subgraph Centrality policy without CSE but can be distinguished by an MPNN + CSE. One graph is a hexagon with a global node connected to all other nodes, and another graph which depicts two triangles connected to a global node.

*Proof.* Let $\{X^i, A^i\}_{i=1}^n$ represent the node feature matrix $X^i$ and adjacency matrix $A^i$, respectively, for the $i$-th subgraph in the collection of subgraphs. We adopt the binary node-marking technique from (Bevilacqua et al., 2021), where $X^i$ includes a one-hot encoded vector for the root node identification, with a 1 in the $i$-th position,[5]

$$X^i = \begin{pmatrix} 0 \\ \vdots \\ 0 \\ 1 \\ 0 \\ \vdots \\ 0 \end{pmatrix} \quad \text{with 1 at the } i\text{-th position}, \tag{14}$$

we note that in this case, the adjacencies of all subgraphs, denoted as $\{A^i\}_{i=1}^n$, are identical to the adjacency of the original graph, $A$.

We recall that the DSS-GNN architecture applies an MPNN over each subgraph independently, followed by an MLP. Subsequently, an MPNN followed by an MLP operates on a shared component, enabling information sharing across subgraphs.

To be more explicit, the architecture is defined as follows:

$$\tilde{X}^{l;i} = \text{MPNN}^{l;1}(A^i, X^{l-1;i}), \tag{15}$$

$$Y^{l-1;i} = f^{l;1}(\tilde{X}^{l-1;i}), \tag{16}$$

$$\tilde{X}^{l-1} = \text{MPNN}^{l;2}\left( \sum_{j=1}^{n} A^j, \sum_{j=1}^{n} X^{l-1;j} \right), \tag{17}$$

$$Y^{l-1} = f^{l;2}(\tilde{X}^{l-1}), \tag{18}$$

$$X^{l+1;i} = Y^{l-1;i} + Y^{l-1}, \tag{19}$$

where the MPNN is defined as:

$$\text{MPNN}(X) = AX\mathbf{W}^1 + X\mathbf{W}^0. \tag{20}$$

The proof proceeds in three steps. First, we compute the centrality encoding at the root nodes, recall Equation (4), specifically at $X_i^i$. Second, we use the shared information component to propagate this root node information across all subgraphs. Finally, we simulate an MPNN over each subgraph. Since the subgraphs are identical and equipped with the centrality encoding, the proof is complete.

**Step 1.** We begin by applying the $K$ MPNN layers to each subgraph independently, and setting to 0 the weight matrices of the shared component, effectively enforcing no sharing between the subgraphs.

More specifically, at the $k$-th layer, the following weight matrices are used:

$$\mathbf{W}^0 = \begin{pmatrix} 1 & 0 & \cdots & 0 & 0 \\ 0 & 1 & \cdots & 0 & 0 \\ \vdots & \vdots & \ddots & \vdots & \vdots \\ 0 & 0 & \cdots & 1 & 0 \end{pmatrix}_{k \times (k+1)} , \quad \mathbf{W}^1 = \begin{pmatrix} 0 & 0 & \cdots & 0 & 0 \\ 0 & 0 & \cdots & 0 & 0 \\ \vdots & \vdots & \ddots & \vdots & \vdots \\ 0 & 0 & \cdots & 0 & 1 \end{pmatrix}_{k \times (k+1)}$$

where:

- $\mathbf{W}^0$ is a $k \times (k+1)$ matrix consisting of a $k \times k$ identity matrix $\mathbf{I}_k$ followed by an extra column of zeros:

$$\mathbf{W}^0 = \begin{pmatrix} \mathbf{I}_k & \mathbf{0}_{k \times 1} \end{pmatrix}$$

---

[5]Although this proof does not consider additional node features, it can be easily adapted to incorporate them.

Specifically, when $\mathbf{W}^0$ is multiplied by a matrix $B \in \mathbb{R}^{n \times k}$, i.e., $B\mathbf{W}^0$, it appends a column of zeros to the right of $B$, leaving the structure of $B$ unchanged but with an additional column of zeros.

- $\mathbf{W}^1$ is a $k \times (k+1)$ matrix consisting of a $k \times k$ zero matrix $\mathbf{0}_{k \times k}$ followed by a column of all zeros except for a 1 in the last row:

$$\mathbf{W}^1 = \begin{pmatrix} \mathbf{0}_{k \times k} & \mathbf{e}_k \end{pmatrix}$$

Specifically, when $\mathbf{W}^1$ is multiplied by a matrix $B \in \mathbb{R}^{n \times k}$, i.e., $B\mathbf{W}^1$, the resulting matrix is composed of the zero matrix and the last column of $B$. In other words, this operation extracts the last column of $B$ and appends it to a matrix of zeros of size $k \times k$.

In this setup, for each $i$, the term $AX^i\mathbf{W}^1$ at the $k$-th layer propagates the node marking to neighboring nodes and places it in the last column. Meanwhile, the term $X\mathbf{W}^0$ copies the propagated marking from the previous $k$ layers.

Thus, by summing the two terms, after $K$ layers, the node features $X_j^i$ are given by:

$$X_j^i = \begin{cases} \left[1, A_{ii}^1, A_{ii}^2, \ldots, A_{ii}^K\right], & \text{for } j = i, \\ \left[0, \mathbf{v}_j\right], & \text{for } j \neq i, \end{cases} \tag{21}$$

where $\mathbf{v}_j$ holds at its $k$-th slot the number of marks propagated to node $j$ in subgraph $i$ at the step $k$.

Since each entry in the vectors $X_j^i$ for any $i \in [n]$ and $j \in [n]$ represents the propagation of a mark over $k \in [K]$ steps — specifically, the number of walks from the root node of subgraph $i$ to node $j$ (within subgraph $i$) — the number of possible values for these vectors is constrained. Moreover, because the original graph is finite, the total number of possible values for these vectors must also be finite.

By Theorem 3.1 in (Yun et al., 2019)[6], there exists an MLP at the $K$-th layer that can implement the following mappings:

$$f^k([1, a_1, a_2, \ldots, a_K]) = \left[1, \frac{a_1}{1!}, \frac{a_2}{2!}, \ldots, \frac{a_K}{K!}\right], \tag{22}$$

$$f^k([z, b_1, b_2, \ldots, b_K]) = \mathbf{0}_{K+1}, \tag{23}$$

for any $z \neq 1$ and $a_i, b_i \in \mathbb{R}$. At this step, the root nodes $i$ hold the centrality value, recall Equation (4), while all other nodes hold the feature vector $\mathbf{0}_{K+1}$.

**Step 2.** Next, we utilize the shared information component by setting the weights of its MPNN as follows, $\mathbf{W}^1 = 0$ and $\mathbf{W}^0 = I$, which effectively broadcasts the root node information to the corresponding nodes in all other subgraphs. To prevent the root nodes from receiving double the value, we initialize the MPNN that operates on each subgraph individually with zero weights.

**Step 3.** At this point, we have $n$ copies of the original graph, each equipped with its corresponding centrality values. Therefore, the MPNN over each subgraph can effectively simulate the MPNN over the original graph, now with centrality values assigned to the nodes. Assuming a mean readout is used at the conclusion of both the DSS-GNN($\mathcal{B}_G$ and the MPNN($G$), their outputs will be identical.

This concludes the proof.

We note that this result is also valid for Subgraph GNN architectures that subsume DSS-GNN, e.g., GNN-SSWL+ (Zhang et al., 2023a). $\square$

---

[6]This theorem assumes the output is bounded between $-1$ and $1$. However, we can relax this assumption as long as the outputs are bounded (and they are since the original graph is finite). To handle this, we can use the theorem to calculate the normalize values (dividing each value by the upper bound), and then use an additional MLP to scale the results back to the original values.

## C ON MARKING-INDUCED PERTURBATIONS AND THE NUMBER OF WALKS

In this section, we report more details and comments on our Observation 1 introduced in Section 3.2.

We start by commenting on Equation (3). We recall that we are interested in upper-bounding the amount of output perturbation induced by marking node $v$. Effectively, this corresponds to the distance $|y_G - y_{S_v}|$, namely, the absolute difference between the predictions a backbone MPNN computes for the original graph ($G$) and the subgraph obtained by marking node $v$ ($S_v$).

For an $L$-layer MPNN in the form of Equation (2), we obtain Equation (3) by an almost immediate application of the results in (Chuang & Jegelka, 2022). Indeed, let us rewrite:

$$|y_G - y_{S_v}| = |\phi^{(L+1)}\big(\sum_{u \in G} h_u^{G,(L)}\big) - \phi^{(L+1)}\big(\sum_{u \in S_v} h_u^{S_v,(L)}\big)| \tag{24}$$

where $h_v^{G,(L)}, h_v^{S_v,(L)}$ indicate, respectively, the representations of node $u$ in graph $G$ and its perturbed counterpart $S_v$. Now, by Chuang & Jegelka (2022, Theorem 8) we have:

$$|y_G - y_{S_v}| \leq \prod_{l=1}^{L+1} K_\phi^{(l)} \cdot \underbrace{\text{TMD}_w^{L+1}(G, S_v)}_{(\mathcal{A})} \tag{25}$$

where $(\mathcal{A})$ is the $L + 1$-depth *Tree Mover's Distance* (TMD) (Chuang & Jegelka, 2022) with layer-weighting $w$ calculated between the original graph and its marked counterpart.

In the same work, the authors provide an upper-bound on the TMD between a graph and a perturbed version obtained by a change in the initial features of a node (Chuang & Jegelka, 2022, Proposition 11). We restate this result.

**Proposition 3.** *(Chuang & Jegelka, 2022, Proposition 11) Let $H$ be a graph and $H'$ be the perturbed version of $H$ obtained by changing the features of node $v$ from $x_v$ to $x_v'$. Then:*

$$TMD_w^L(H, H') \leq \sum_{l=1}^{L} \lambda_l \cdot Width_l(T_v^L) \cdot \|x_v - x_v'\| \tag{26}$$

*where $\lambda_l \in \mathbb{R}^+$ is a layer-wise weighting scheme dependent of $w$ and $Width_l(T_v^L)$ is the width at the $l$-th level of the $L$-deep computational tree rooted in $v$.*

We can readily apply Proposition 3 and leverage the fact that marking only induces a unit-norm feature perturbation to get:

$$|y_G - y_{S_v}| \leq \prod_{l=1}^{L+1} K_\phi^{(l)} \cdot \text{TMD}_w^{L+1}(G, S_v) \tag{27}$$

$$\leq \prod_{l=1}^{L+1} K_\phi^{(l)} \cdot \sum_{l=1}^{L+1} \lambda_l \cdot \text{Width}_l(T_v^{L+1}) \tag{28}$$

Finally, we note that $\text{Width}_l(T_v^{L+1})$ corresponds to the number of walks of length $l - 1$ starting from $v$. This can be easily seen by noting that the leaves of the computational tree can be put in a bijection with all and only those walks of length $l - 1$ starting from $v$[7]. This value is notoriously computed from row-summing powers of the adjacency matrix $A$, so that: $\text{Width}_l(T_v^{L+1}) = \sum_j (A^{l-1})_{v,j}$. We ultimately have:

$$|y_G - y_{S_v}| \leq \prod_{l=1}^{L+1} K_\phi^{(l)} \cdot \sum_{l=1}^{L+1} \lambda_l \cdot \text{Width}_l(T_v^{L+1}) \tag{29}$$

$$= \prod_{l=1}^{L+1} K_\phi^{(l)} \cdot \sum_{l=1}^{L+1} \lambda_l \cdot \sum_j (A^{l-1})_{v,j} \tag{30}$$

---

[7]This can be constructed, e.g., by associating leaf nodes to the walks (uniquely) obtained by "climbing up" the computational tree up to the root.

The above bound is easily extended to consider a closely related analysis: the impact of adding a single node-marked subgraph $S$ to a bag formed by the original graph only. This analysis would enlighten us on the impact a single subgraph addition in the case of "augmented policies", which we use throughout our experiments in Section 5, see Appendix E.1.

In other words, we would like to bound $|y_G^{B=\{G\}} - y_G^{B=\{S,G\}}|$, where these outputs are given by Equation (39), with a base MPNN backbone as per Equation (2). We have:

$$|y_G^{B=\{G\}} - y_G^{B=\{S,G\}}| = |\phi^{(L+1)}\big(\sum_{v\in G}(h_{G,v}^{(L)} + h_{S,v}^{(L)})\big) - \phi^{(L+1)}\big(\sum_{v\in G} h_{G,v}^{(L)}\big)| \qquad (31)$$

$$\leq K_\phi^{L+1} \cdot \|\sum_{v\in G}(h_{G,v}^{(L)} + h_{S,v}^{(L)}) - \sum_{v\in G} h_{G,v}^{(L)}\| \qquad (32)$$

where $K_\phi^{L+1}$ is the Lipschitz constant of the prediction layer $\phi^{(L+1)}$. We can rewrite the above as follows by appropriately rearranging terms and by the triangular inequality:

$$|y_G^{B=\{G\}} - y_G^{B=\{S,G\}}| \leq K_\phi^{L+1} \cdot \|\sum_{v\in G}(h_{G,v}^{(L)} + h_{S,v}^{(L)}) - \sum_{v\in G} h_{G,v}^{(L)}\| \qquad (33)$$

$$\leq K_\phi^{(L+1)} \cdot \Big(\|\sum_{v\in G} h_{G,v}^{(L)}\| + \underbrace{\|\sum_{v\in G} h_{S,v}^{(L)} - \sum_{v\in G} h_{G,v}^{(L)}\|}_{(\mathcal{A})}\Big) \qquad (34)$$

where, we note, $(\mathcal{A})$ is the distance between the embeddings of the marked and unmarked graphs, before a final predictor is applied. This term can be bounded similar to our initial analysis for Observation 1:

$$(1) = \|\sum_{v\in G} h_{S,v}^{(L)} - \sum_{v\in G} h_{G,v}^{(L)}\| \qquad (35)$$

$$\leq \prod_{l=1}^{L} K_\phi^{(l)} \cdot \underbrace{\text{TMD}_w^{L+1}(S,G)}_{(\mathcal{B})} \qquad (36)$$

where $(\mathcal{B})$ can be upper-bounded, again, by Proposition 3[8]. Putting things together:

$$|y_G^{B=\{G\}} - y_G^{B=\{S,G\}}| \leq K_\phi^{(L+1)} \cdot \Big(\|\sum_{v\in G} h_{G,v}^{(L)}\| + \prod_{l=1}^{L} K_\phi^{(l)} \cdot \sum_{l=1}^{L+1} \lambda_l \cdot \sum_j (A^{l-1})_{S,j}\Big) \qquad (37)$$

Differently from the above analysis, we observe a contribution given by $\|\sum_{v\in G} h_{G,v}^{(L)}\|$. This could be upper-bounded, e.g., by the sum of the "tree-norms" Chuang & Jegelka (2022) of the computational trees over the original graph. We note that (the presence of) this term is, however, independent on the selection of the specific node to mark.

In future developments of this work we envision to more deeply enquire into the relation between Equation (37) and Equation (1), and into the principled choice of a specific centrality measure among different possibilities.

# D ADDITIONAL EXPERIMENTS

## D.1 COMPARISON TO OTHER CENTRALITY MEASURES

We compared the impact of using different node centrality measures as a node marking scheme on how much they altered the graph representation. For each of these centrality measures, we measured the distance $\|f(S_v) - f(G)\|$ on 100 graphs from two different real-world datasets from the popular

---

[8]We have allowed a little abuse of notation here by using $S$ to refer to the "subgraph" obtained by marking node $S$ in $G$.

TU suite: MUTAG and NCI1 (Morris et al., 2020a). Here, $f$ is an untrained 3-layer GIN (Xu et al., 2018). We can see from Table 5 that marking the node with using the maximum values of the three walk-based centrality measures (Subgraph, Communicability, Katz) leads to the highest average perturbation and marking the node with the minimum of these centrality measures leads to the lowest. This implies that this family of centrality measures is most aligned with the perturbation distance.

*Table 5:* Amount of perturbation from the original graph representation on MUTAG and NCI1 using 3-layer untrained GIN with 32 hidden dimension by incorporating a node-marked subgraph with different marking policies.

| Marking Policy | MUTAG Perturbation | NCI1 Perturbation |
|---|---|---|
| Random | 0.0648 | 0.0126 |
| Minimum Degree Centrality | 0.0202 | 0.0075 |
| Maximum Degree Centrality | 0.0968 | 0.0075 |
| Minimum Closeness Centrality | 0.0241 | 0.0073 |
| Maximum Closeness Centrality | 0.1038 | 0.0184 |
| Minimum Betweenness Centrality | 0.0202 | 0.0076 |
| Maximum Betweenness Centrality | 0.0957 | 0.0183 |
| Minimum Katz Centrality | 0.0177 | 0.0063 |
| Maximum Katz Centrality | 0.1051 | 0.0200 |
| Minimum Communicability Centrality | 0.0177 | 0.0063 |
| Maximum Communicability Centrality | 0.1056 | 0.0200 |
| Minimum Subgraph Centrality | 0.0177 | 0.0063 |
| Maximum Subgraph Centrality | 0.1055 | 0.0201 |

To further assess the benefits of our specific centrality encoding for sampling subgraphs, we compared against using other centrality measures to sample subgraphs in the counting substructure task. We used Closeness centrality, Betweeness centrality, Pagerank centrality and Degree centrality as baselines using the Networkx library (Hagberg et al., 2008). From Fig. 6, we can see that using any of the different centrality methods performs better than random sampling across all substructures and number of samples. We also find that the Subgraph Centrality which we use, outperforms all other approaches in counting 3 and 4-cycles for any number of samples and in counting 3 and 4-paths when number of samples $\geq 5$. For 4-cycles and other substructures, we find that Subgraph centrality is best, followed by Degree and Pagerank centrality and then Closeness and Betweenness centralities perform the worst of these centrality measures. This ranking of performance is aligned with how correlated these substructures are with the Subgaph Centrality on these synthetic graphs (as shown in Table 6).

To further compare different centrality measures, we ran additional experiments on the Peptides and MolHIV datasets. We experimented in particular, with the Betweenness Centrality (BC), the Katz Index (KI) and the Subgraph Centrality (SC). We see from Table 7 that the performances achieved by different centrality measures are not dramatically different from each other, with those by the KI and SC being closer. In fact, centrality measures often exhibit a degree of correlation with each other, especially if from the same family, as it is the case of the walk-based KI and SC (see Estrada & Rodriguez-Velazquez (2005) and Table 6). It is also worth noting that Subgraph Centrality can be more efficient to calculate than these other centrality measures using the Networkx library (see Table 8).

Overall, we believe that specific centrality measures could work better than others depending on the task at hand, but, at the same time, our current ensemble of observations indicate that walk-based centrality measures – and, in particular, the Subgraph Centrality – offer the most competitive results for the lightest precomputation run-time. Given the additional support provided by the bound discussed in Section 3, we think they constitute particularly strong candidates across use-cases.

## D.2 COMPARISON BETWEEN CENTRALITY-BASED STRUCTURAL ENCODINGS AND RWSE

Here we aim to outline some of the similarities and differences between our Centrality structural Encoding (CSE) defined in Eq. (4) and the Random-Walk Structural Encoding (RWSE) introduced in (Dwivedi et al., 2021). The RWSE uses the diagonal of the $k$-step random-walk matrix defined in

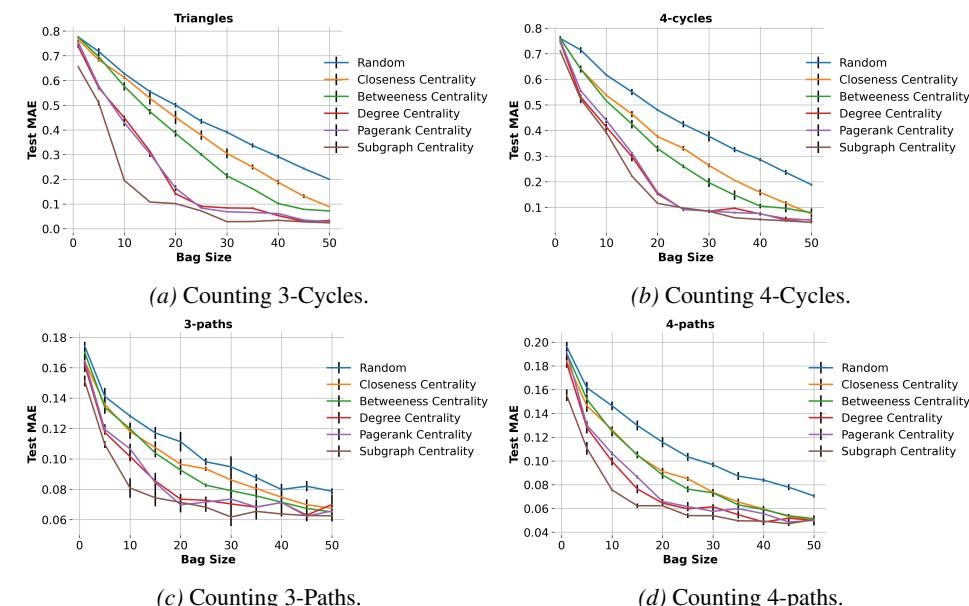

*(a)* Counting 3-Cycles.

*(b)* Counting 4-Cycles.

*(c)* Counting 3-Paths.

*(d)* Counting 4-paths.

*Figure 6:* Comparing different centrality measures for counting different substructures on synthetic random graphs.

*Table 6:* Correlation between the different centralities and Subgraph Centrality on the random regular graphs in the substructure counting experiments.

| Centrality | Correlation with Subgraph Centrality |
|---|---|
| Pagerank Centrality | $0.923 \pm 0.025$ |
| Degree Centrality | $0.970 \pm 0.012$ |
| Betweeness Centrality | $0.801 \pm 0.074$ |
| Closeness Centrality | $0.786 \pm 0.067$ |

Eq. (38) defined as:

$$p_i^{\mathsf{RWSE}} = [(AD^{-1})_{ii}, (AD^{-1})_{ii}^2, \ldots, (AD^{-1})_{ii}^k] \in \mathbb{R}^k, \tag{38}$$

These terms show similarity to our CSE as it also stores powers of the diagonal of the adjacency matrix, but it has a different normalization term that depends on the degree. In Proposition 1, we show that using an MPNN with CSE can compute the the probability, for all possible walks departing from a node, that a walk will lead back to the start. RWSE structural encodings are subtly different in that they compute the landing probability of a *random* walk from a node to itself. In this case, rather than weighting all possible walks equally, the walks are weighted by the degrees of the nodes along the walk; a walk where there are fewer alternative routes (other nodes) for the RW is more likely to occur.

Here we aim to empirically examine this difference to see (i) which one is a more effective Structural Encoding and (ii) which one is more effective for subgraph sampling. To show the effect of both CSE and RWSE as SEs, we compared both on the Peptides datasets (Dwivedi et al., 2022) with two different base MPNNs (GCN and GIN). From Table 9, it can be seen that our centrality encoding performs similarly to the RWSE encoding; matching almost exactly except with a GCN on Peptides-Func.

To answer (ii) and highlight the benefit of sampling based on CSE over using RWSE, we compared using the sum of these different encodings to sample the subgraphs in the counting substructures experiment. From Fig. 7, we see that our sampling method is better for counting all substructures and for all the number of samples in comparison to RWSE sampling.

In conclusion, Appendix E.2 shows that our CSE is better for sampling subgraphs and Table 9 shows that CSE is competitive when purely used as a Structural Encoding. Therefore, it is well motivated to

*Table 7:* Comparison of different centrality measures on real-world molecular datasets.

| Centrality for Sampling | MolHIV | Peptides-Func | Peptides-Struct |
|---|---|---|---|
| Betweenness Centrality | 78.86 ±0.98 | 0.6749 ±0.0066 | 0.2478 ±0.0006 |
| Katz Index | 79.58 ±0.98 | 0.6756 ±0.0056 | 0.2469 ±0.0008 |
| Subgraph Centrality | 79.77 ±0.70 | 0.6758 ±0.0050 | 0.2466 ±0.0010 |

*Table 8:* Timing of different centrality measures on an Erdös-Renyi graph with 1000 nodes and p=0.5 using the Networkx library.

| Centrality | Time (s) |
|---|---|
| Betweenness Centrality | 83.12 |
| Katz Index | 1.31 |
| Subgraph Centrality | 0.54 |

use the CSE for our hybrid method where we need an SE *and* to use it as a sampling method. Future work could consider further understanding the expressivity differences between these SEs and the role of the normalization factor.

### D.3 FURTHER EXAMINING THE EFFECT OF CSEs

In Table 2, we explore the effect of HyMN with and without CSEs. In order to complement these results, we additionally evaluated the effect of CSEs on HyMN and GIN with the Peptides and ZINC datasets. The results are reported in Table 10. These results further show that adding even one subgraph with our approach can be beneficial and that additionally using the centrality measure as a structural encoding can also improve performance.

## E EXPERIMENTAL DETAILS

In this section we provide details on the experimental validation described and discussed in Section 5.

### E.1 ARCHITECTURAL FORM

We always employ a reference Subgraph GNN architecture $f$ whose output, for an input graph $G = (A, X, E)$[9] associated with node-marked bag $B$, is given by:

$$y_G^B = f(B(G)) = \phi^{(L+1)}\Big(\sum_{v \in G}(h_{G,v}^{(L)} + \sum_{S \in B} h_{S,v}^{(L)})\Big) \tag{39}$$

$$h_{S,v}^{(l)} = \mu^{(l)}\big(A, H_S^{(l-1)}, \eta_e(E), M_{:,S}\big)_v \tag{40}$$

$$h_{S,v}^{(0)} = [\eta_x(X)_{v,:}, C_{v,:}^{\mathsf{CSE}}] \tag{41}$$

where $\phi^{(L+1)}$ is a final prediction module and $h_{G,v}^{(L)}, h_{S,v}^{(L)}$ refer to the representations of generic node $v$ on the original graph $G$ and subgraph $S$. As it is evident from Equation (39), we employ an "augmented policy" which always includes a copy of the original graph in the bag of

---

[9]$E$ is a tensor storing edge features.

*Table 9:* Results on the Peptides datasets comparing CSE with RWSE.

| Method | Peptides-Func ($\uparrow$) | Peptides-Struct ($\downarrow$) |
|---|---|---|
| GIN | 0.6555 ±0.0088 | 0.2497 ±0.0012 |
| GIN + RWSE | 0.6621 ±0.0067 | 0.2478 ±0.0017 |
| GIN + CSE | 0.6619 ±0.0077 | 0.2479 ±0.0011 |
| GCN | 0.6739 ±0.0024 | 0.2505 ±0.0023 |
| GCN + RWSE | 0.6860 ±0.0050 | 0.2498 ±0.0015 |
| GCN + CSE | 0.6812 ±0.0037 | 0.2499 ±0.0010 |

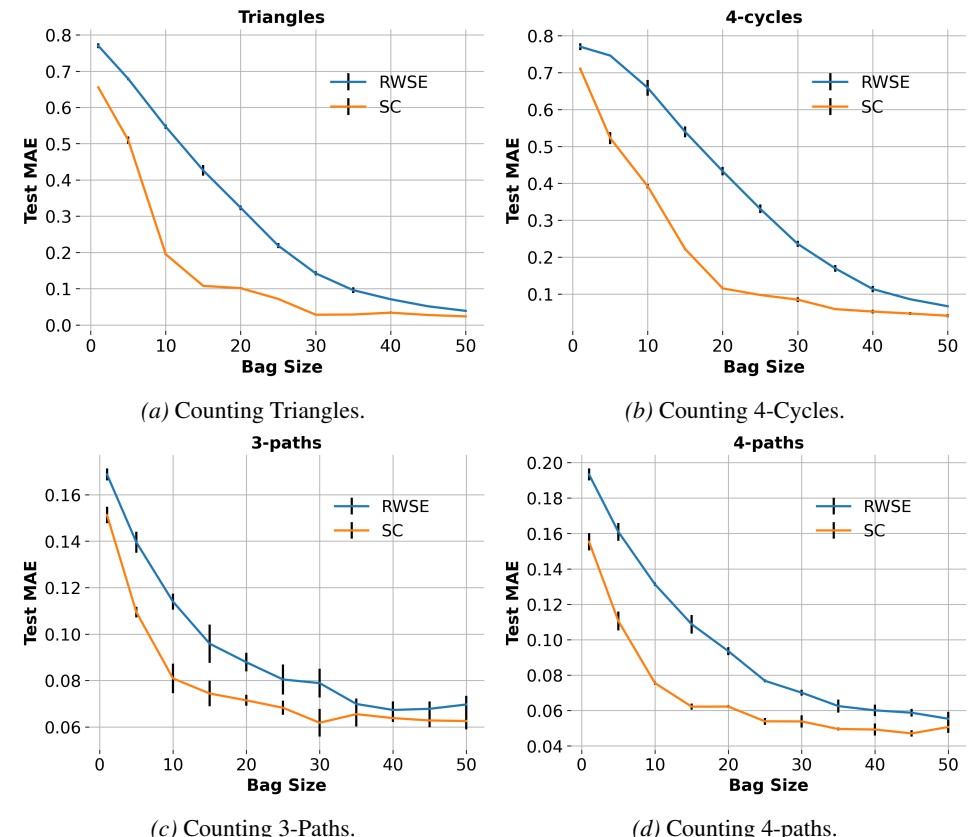

*(a)* Counting Triangles.          *(b)* Counting 4-Cycles.

*(c)* Counting 3-Paths.          *(d)* Counting 4-paths.

*Figure 7:* Comparing our centrality sampling with RWSE sampling for counting different substructures on synthetic random graphs.

*Table 10:* Results on ZINC and Peptides datasets showing the effect of CSEs on both GIN and HyMN.

| Method | ZINC (MAE ↓) | Peptides-Func (AP ↑) | Peptides-Struct (MAE ↓) |
|---|---|---|---|
| GIN Xu et al. (2018) | 0.163 ±0.004 | 0.6558 ±0.0068 | 0.2497 ±0.0012 |
| GIN+CSE | 0.092 ±0.002 | 0.6619 ±0.0077 | 0.2479 ±0.0011 |
| **HyMN (GIN, T=1) w/out CSE** | 0.125 ±0.004 | 0.6758 ±0.0050 | 0.2466 ±0.0010 |
| **HyMN (GIN, T=1)** | 0.080 ±0.003 | 0.6857 ±0.0055 | 0.2464 ±0.0013 |

subgraphs (Bevilacqua et al., 2021). As we only consider node marking policies, this copy only differs from (sub)graphs in $B$ by the fact that no nodes are marked. Representations $h_{G,v}^{(L)}, h_{S,v}^{(L)}$ are obtained à la DS-GNN (Bevilacqua et al., 2021), that is, by running independent message-passing $\mu$ on each (sub)graph independently (see Equation (40)). Note that $\mu$ explicitly processes available edge features ($E$) and marking information ($M$, see Algorithm 1). As it will be specified later on, we always consider either GIN (Xu et al., 2018) or GCN (Kipf & Welling, 2017) as MPNN backbones. Equation (41) specifies initial node features for node $v$ in subgraph $S$. Finally, $\eta_x$, $\eta_e$ are dataset-dependent node- and edge-feature encoders, and that $C^{\mathsf{CSE}}$ is computed according to Algorithm 1.

Note that, across all molecular benchmarks, the GIN layer we use resembles the GINE architecture (Hu et al., 2020c), but concatenates the marking information as follows:

$$h_{S,v}^{(l)} = \phi^{(l)}\Big((1 + \epsilon^{(l)})[h_{S,v}^{(l-1)}, M_{v,S}] + \sum_{u \in N(v)} \big[\sigma\big(h_{S,u}^{(l-1)} + \eta_e(E)_{vu}\big), M_{u,S}\big]\Big) \qquad (42)$$

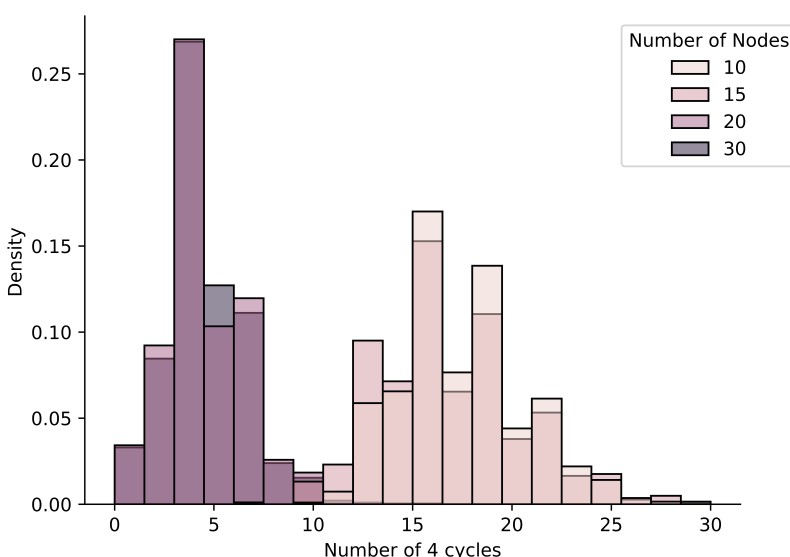

*Figure 8:* Evaluating the dependency between the graph size and the number of 4-cycles in the dataset generated from Chen et al. (Chen et al., 2020).

where $M_{v,S}$ is the mark for node $v$ in subgraph $S$, $\eta_e(E)_{vu}$ refers to the encoded edge features for node-pair $v, u$, and $\sigma$ is a ReLU non-linearity. As for our GCN (Kipf & Welling, 2017) backbones, the marking information is simply provided in the input of the network, and edge features are discarded.

## E.2 SYNTHETIC EXPERIMENTAL DETAILS

### E.2.1 DATASET GENERATION

For the synthetic counting substructures experiment, we generated a dataset of random unattributed graphs in a similar manner to (Chen et al., 2020). In their experiments, they generate 5000 random regular graphs denoted as $RG(m, d)$, where $m$ is the number of nodes in each graph and $d$ is the node degree. Random regular graphs with $m$ nodes and degree $d$ are sampled and then $m$ edges are randomly deleted. In their work, Chen et al. (Chen et al., 2020), uniformly sampled $(m, d)$ from $(10, 6), (15, 6), (20, 5), (30, 5)$. However, we want to test the effectiveness of our sampling approach for larger sizes of graphs. Additionally, we found that the number of substructures present in the graph was related to the graphs size (see Fig. 8). Therefore, we wanted to create a more challenging benchmark with larger graph sizes. Therefore, we set $(m, d)$ to be $(60, 5)$ for all graphs.

### E.2.2 SYNTHETIC MODEL PARAMETERS

For our synthetic experiments, we set the base GIN to have a batch size of 128, 6 layers of message-passing, embedding dimension 32, and Adam optimizer with initial learning rate of 0.001 as prescribed by (Bevilacqua et al., 2024). We trained for 250 epoch and took the test Mean Absolute Error (MAE) at the best validation epoch.

## E.3 REAL-WORLD EXPERIMENTAL DETAILS

In this section, we provide further details about our experiments. We implemented our method using Pytorch (Paszke et al., 2019) and Pytorch Geometric (Fey & Lenssen, 2019). For the GIN model (Xu et al., 2018), we use Batch Normalization and the MLP is composed of two linear layers with ReLU non-linearities. Additionally, we use residual connections in each layer. The test performance at the epoch with the best validation performance is reported and is averaged over multiple runs with different random seeds. All the benchmarking results, including the extra ablations, are based on 5

executed runs, except for Peptides-func and Peptides-struct which are based on the output of four runs. In all our experiments we used AdamW (Loshchilov & Hutter, 2019), together with linear warm-up increase of the learning rate followed by its cosine decay. Experimental tracking and hyper-parameter optimisation were done via the Weights and Biases platform (wandb) (Biewald et al., 2020). In Table 2, the number of subgraphs used ($T$) was selected to match the choices made in other baselines, such as PL, making the comparisons as informative as possible. In Table 3 and Table 4, we chose $T$ to be the smallest possible value, i.e., $T = 1$. This is justified by our focus on efficiency. Specific hyper-parameter information for each dataset can be found in the corresponding subsection.

### E.3.1 HARDWARE

All experiments were run on a single NVIDIA GeForce RTX 3080 with 10GB RAM.

### E.3.2 DATASET SPECIFIC DETAILS

Below, we provide descriptions of the datasets on which we conduct experiments.

**OGB datasets** (MIT License) (Hu et al., 2020b). These are molecular property prediction datasets which use a common node and edge featurization that represents chemophysical properties. MOL-HIV, MOLBACE and MOLTOX21 all represent molecule classification tasks. We considered the challenging scaffold splits proposed in (Hu et al., 2020a). We set the batch size to 128 for MOLHIV and 32 for the other benchmarks to avoid out-of-memory errors. We set the hidden dimension to be 300 for all datasets as done in Hu et al. (2020a) and Bevilacqua et al. (2024). We tuned the number of layers in $2, 4, 6, 8, 10$, the number of layers post message-passing in $1, 2, 3$, dropout after each layer in $0.0, 0.3, 0.5$, whether to perform mean or sum pooling over the subgraphs, and whether to apply Batch Normalization after message-passing on each dataset. Additionally, for the method with structural encoding, we tune the number of steps $k$ in the encoding in $16, 20$ and the dimension after the linear encoding in $16, 28$ as done in (Rampášek et al., 2022). The tuning was done on a single run for each set of hyper-parameters and the results were outlined for the best performing parameters on the validation set over 5 random seeds. These parameters are shown in Table 11.

The maximum number of epochs is set to 100 for all models and the test metric is computed at the best validation epoch.

*Table 11:* Best performing hyperparameters in Table 2.

| Hyperparameter | MOLHIV | MOLBACE | MOLTOX21 |
|---|---|---|---|
| #Layers | 2 | 8 | 10 |
| #Layers readout | 1 | 3 | 3 |
| Hidden dim | 300 | 300 | 300 |
| Dropout | 0.0 | 0.5 | 0.3 |
| Subgraph pooling | mean | mean | sum |
| Positional Encoding Steps | 16 | 20 | 20 |
| PE dim | 16 | 16 | 28 |
| #Parameters | 419,403 | 1,691,329 | 2,061,322 |

**Peptides-func and Peptides-struct** (CC-BY-NC 4.0) (Dwivedi et al., 2022). These datasets are composed of atomic peptides. Peptides-func is a multi-label graph classification task where there are 10 nonexclusive peptide functional classes. Peptides-struct is a regression task involving 11 3D structural properties of the peptides. For both of these datasets, we used the tuned hyper-parameters of the GINE model from Tönshoff et al. (Tönshoff et al., 2023) which has a parameter budget under 500k and where they use 250 epochs. For both of these datasets we set the number of steps of our centrality encoding to be 20, aligned with the number of steps used for the random-walk structural encoding. The additional parameter tuning which we performed was whether to do mean or sum pooling over the subgraphs. We show the best performing hyperparameters from Table 3 in Table 12.

**ZINC** (MIT License) (Dwivedi et al., 2023). This dataset consists of 12k molecular graphs representing commercially available chemical compounds. The task involves predicting the constrained solubility of the molecule. We considered the predefined dataset splits and used the Mean Absolute Error (MAE) both as a loss and evaluaton metric. We chose to have 10 layers of massage-passing, 3 layers in the readout function, a batch size of 32, 1000 epochs and a dropout of 0 to replicate what

*Table 12:* Best performing hyperparameters in Table 3.

| Hyperparameter | Peptides-Func | Peptides-Struct |
|---|---|---|
| #Layers | 8 | 10 |
| #Layers readout | 3 | 3 |
| Hidden dim | 160 | 145 |
| Dropout | 0.1 | 0.2 |
| Subgraph pooling | sum | sum |
| Positional Encoding Steps | 20 | 20 |
| PE dim | 18 | 18 |
| #Parameters | 498,904 | 496,107 |

was done in (Rampášek et al., 2022). We altered the hidden dimension to be 148 in order to be closer to the 500k parameter budget. No further parameter tuning was done and our best performing parameters are shown in Table 13. The test metric is computed at the best validation epoch.

*Table 13:* Best performing hyperparameters in Table 4.

| Hyperparameter | ZINC |
|---|---|
| #Layers | 10 |
| #Layers readout | 3 |
| Hidden dim | 148 |
| Dropout | 0.0 |
| Subgraph pooling | mean |
| Positional Encoding Steps | 20 |
| PE dim | 18 |
| #Parameters | 497,353 |

# F    MORE ON SUBGRAPH GNNS AND THEIR COMPLEXITY

## F.1    THE ARCHITECTURAL FAMILY OF SUBGRAPH GNNS

The term "Subgraph GNN" refers to a broad family of recent Graph Neural Networks sharing a common architectural pattern: that of modeling graphs as sets (bags) of subgraphs. Subgraphs are processed by a *backbone* GNN, possibly flanked by additional information sharing modules (Bevilacqua et al., 2021). Bags of subgraphs are formed by *selection policies*, which typically extract subgraphs by applying topological perturbations such as node- (Cotta et al., 2021; Papp et al., 2021) or edge-deletions (Bevilacqua et al., 2021), or by marking nodes (You et al., 2021; Papp & Wattenhofer, 2022).

In formulae, a Subgraph GNN $f$ can be described as (Frasca et al., 2022):

$$f : G \mapsto \big(\mu \circ \rho \circ \mathcal{S} \circ \pi\big)(G), \tag{43}$$

where $\pi$ is the selection policy; $\mathcal{S}$ applies the backbone GNN – with, potentially, information sharing components; $\rho, \mu$ are pooling and prediction modules.

Various choices for the above terms give rise to different Subgraph GNN variants (Frasca et al., 2022). For non-trivial selection policies and sufficiently expressive backbones, these exceed 1-WL discriminative power (Bevilacqua et al., 2021), thus surpassing standard message-passing networks.

The most popular selection policies are *node-based*: selected subgraphs are in a bijection with nodes in the original input graph. Prominent policies in this class include node-deletion, ego-networks and node-marking. From an expressiveness perspective, node-marking subsumes the first two policies (Papp & Wattenhofer, 2022; Zhang et al., 2023a). This policy constructs bags of subgraphs as:

$$\pi_{\mathsf{NM}} : G = (A, X) \mapsto \{(A, X \oplus e_1), \dots, (A, X \oplus e_n)\}, \tag{44}$$

where $e_i \in \{0,1\}^{n \times 1}$ is $i$-th element of the canonical basis [10] and $\oplus$ denotes concatenation across the channel dimension.

---

[10]Elements in vector $e_i$ are 0 except for the one in position $i$, which equals 1.

Node-based Subgraph GNNs encompass several architectures, including ID-GNNs (You et al., 2021), $(n-1)$-Reconstruction GNNs (Cotta et al., 2021), Nested GNNs (Zhang & Li, 2021), GNN-AK (Zhao et al., 2022), SUN (Frasca et al., 2022) and the maximally expressive GNN-SSWL (Zhang et al., 2023a). For 1-WL-expressive backbones, Subgraph GNNs with node-based policies are bounded in their expressive power by 3-WL (Frasca et al., 2022). Detailed, structured charting of their design space, along with fine-grained expressiveness results, are found in (Frasca et al., 2022; Zhang et al., 2023a).

### F.2 COMPUTATIONAL COMPLEXITY ASPECTS

Consider a Subgraph GNN $f$ in the form of Equation (43), where $\mathcal{S}$ stacks neural message-passing layers. For an input graph $G$ with $n$ nodes and a degree bounded by $d_{\max}$, $f$ exhibits an asymptotic forward-pass complexity:

$$T(n, d_{\max}, m) = \mathcal{O}(m \cdot \overbrace{n \cdot d_{\max}}^{\text{msg-pass complexity}}), \tag{45}$$

with $m$ being the number of subgraphs generated by policy $\pi$ executed on graph $G$.

Node-based policies are such that the number of subgraphs equals the number of nodes in the original input graph, i.e., $m = n$. Hence, the complexity of node-based Subgraph GNNs scales as:

$$T(n, d_{\max}) = \mathcal{O}(n^2 \cdot d_{\max}). \tag{46}$$

Higher-order policies (Qian et al., 2022) may allow larger expressive power, but for heftier complexities. As an example, node-pair marking would induce a complexity of $\mathcal{O}(n^3 \cdot d_{\max})$.

The quadratic dependency on the number of nodes in Equation (46) hinders the application of Subgraph GNNs to even mildly-sized graphs. At the time of writing, experimenting on the Peptides datasets (Dwivedi et al., 2022) is already very challenging on common hardware, despite these graphs having on average $\approx 151$ nodes and a similar number of edges.

### F.3 SUBSAMPLING BAGS OF SUBGRAPHS

To mitigate the aforementioned issue, one possibility is to reduce the number of subgraphs to process, i.e., to lower the impact of $m$ in Equation (45). A convenient approach is to *sample* a small set $k$ of subgraphs from the bag generated by a predefined policy $\pi$ (Bevilacqua et al., 2021; Zhao et al., 2022; Bevilacqua et al., 2024; Kong et al., 2024; Sun et al., 2021). Essentially, this requires updating Equation (43) as:

$$f : G \mapsto \left(\mu \circ \rho \circ \mathcal{S} \circ \overbrace{\sigma}^{\text{sampling}} \circ \pi\right)(G). \tag{47}$$

Above, $\sigma$ applies a subgraph sampling strategy to reduce the bag cardinality from $m$ to $k$:

$$\sigma : B = \{G_1, \ldots, G_m\} \mapsto \tilde{B} \quad \text{s.t.} \quad \tilde{B} \subseteq B, \ |\tilde{B}| = k. \tag{48}$$

The new cardinality $k$ should scale sub-linearly in $n$, or be chosen as an appropriate small constant. The optimal design of strategy $\sigma$ is a non-trivial task at the core of several recent works (Qian et al., 2022; Kong et al., 2024; Bevilacqua et al., 2024; Bar-Shalom et al., 2024b). In the present manuscript, we discuss a simple, effective construction based on walk-based centrality measures. Our design is justified by the theoretical considerations and empirical observations in Section 3.2, which are validated by the complementary experimental results reported in Section 5.

## G ADDITIONAL TIME COMPARISONS

As well as Table 3 where we compared the runtime of different methods on Peptides datasets, we extend this analyses to both ZINC and MOLHIV. To this end, we provide some results in Table 14 and Table 15. We find that our method significantly improves over the baseline GIN on both of these tasks whilst having a substantially reduced runtime compared to the GPS which uses a Transformer layer. Again, this highlights both the efficiency and practical utility of our method.

*Table 14:* Results on the ZINC dataset with timing comparisons using a GeForce RTX 2080 8 GB.

| Method | Precompute (s) | Train (s/epoch) | Test (s) | Test MAE (↓) |
|---|---|---|---|---|
| GIN | 0.00 | 12.65 | 0.33 | 0.163 |
| **HyMN (GIN, T=1)** | 21.41 | 17.95 | 0.42 | 0.080 |
| GPS (Rampášek et al. (2022)) | 19.13 | 33.02 | 0.87 | 0.070 |

*Table 15:* Results on the MOLHIV dataset with timing comparisons using a GeForce RTX 2080 8 GB.

| Method | Precompute (s) | Train (s/epoch) | Test (s) | ROC-AUC (↑) |
|---|---|---|---|---|
| GIN | 0.00 | 7.50 | 0.33 | 75.58 |
| **HyMN (GIN, T=1)** | 67.43 | 9.09 | 0.37 | 80.36 |
| GPS (Rampášek et al. (2022)) | 40.92 | 124.08 | 4.14 | 78.80 |

