# OpenReview forum: "Balancing Efficiency and Expressiveness: Subgraph GNNs with Walk-Based Centrality"
_ICLR.cc/2025/Conference — Submitted to ICLR 2025_

### Official Review · Reviewer_pVQU · 2024-10-30

**Soundness:** 2
**Presentation:** 3
**Contribution:** 2
**Rating:** 6
**Confidence:** 3

**Summary:**

The proposed work mainly addresses concerns regarding the temporal complexity of Subgraph GNNs. It proposes a mechanism based on subgraph centrality to sample the graphs that will be part of the bag. Furthermore, it demonstrates that adding centrality encoding to nodes can enhance the discriminative power of sampled Subgraph GNNs while limiting the added computational overhead.

Main Contributions:
1. Adoption of centrality-based sampling for Subgraph GNNs.
2. Clear statement and results regarding incorporating structural encoding as means to improve expressivity of sampled Subgraph GNNs without much computational overhead.

**Strengths:**

1. **Relevance and Contextualization**: The proposed work addresses an issue (computational complexity of Subgraph GNNs) that is extremely relevant for the community. It clearly identifies gaps in existing methods and situates the research well within the current state of the art.
2. **Centrality-based Sampling**: The decision of sampling based on centrality measures has some theoretical backing. The analysis based on perturbations is well founded and creative.
3. **Structural Encoding**: The decision of incorporating structural encodings with subsampling is well motivated and has sufficient theoretical backing.

**Weaknesses:**

1. **Theoretical Concerns**: The analysis based on perturbation is not sufficient to justify using the highest centrality nodes to guide the sampling procedure. Even though they will tend to produce the highest perturbation in the graph representation, this alone does not fully justify their selection for the sampling procedure. These nodes may cause the highest perturbation in graph representations, but their practical value in enhancing graph learning remains uncertain, there is no guarantee on the usefulness and attainability of such modifications. Additionally, the assumption that high-centrality nodes capture the most relevant information is limited by the specific centrality metric used, which may not capture all essential features.
2. **Comparison of Centrality Measures**: The comparison between the adopted subgraph centrality, SC, and other centralities is not sufficient to prove SC's superiority. Moreover, centrality measures often complement each other rather than subsumming one another. SC may be more effective for certain tasks but not necessarily for all graph learning tasks. Rather than claiming SC's superiority, it would be better to show in which tasks SC excels and acknowledge that other centrality measures may outperform it in different contexts. Otherwise, stronger results linking the superiority of SC over every other centrality for graph learning, would be necessary.
3. **RWSE vs CSE**: While the proposed method (CSE) shows some advantages over RWSE, particularly from a sampling perspective as seen in Figure 7, its benefits appear limited. The experiments focus primarily on counting substructures, a relevant task but one that may not fully demonstrate CSE's broader applicability due to its inherent predisposition toward this type of task.
4. **Inconsistencies in Experiments**: The experimental results lack consistency, as the models used for comparison vary across datasets. It is understandable that in many cases the results for all datasets are not available in the original works. However, this inconsistency can raise confusion and concerns of cherry-picking. It would strengthen the results to ensure uniformity in model comparisons.
5. **Missing Runtime Comparisons**:  The efficiency of the proposed method is emphasized throughout the paper, yet runtime comparisons are not provided for all datasets. Since computational efficiency is a key focus, these comparisons should be included in every experiment to give a more comprehensive view of the method’s benefits.
6. **Confusing Proof**: The presentation of Theorem 4 lacks clarity. Specifically, in line 910, it is stated that Lemma 1 will be used to demonstrate that the multiset of values in the centrality encoding for each graph will be distinct. However, unless I am overlooking something, Lemma 1 does not seem to establish this. Rather, it appears to indicate that the global node is consistently selected for top-1 sampling, which does not sufficiently ensure distinguishability between the centrality multisets of the two graphs. This interpretation seems supported by the statement in line 963. Moreover, the topic of centrality encoding does not reappear until line 966.

Furthermore, considering that $A$ represents the adjacency matrix and $A^k$ denotes its $k$th power, the assertion in Equation 11 raises concerns regarding its mathematical validity and intuitive clarity. For example, the expression $A^{k+1}\_{u\_1,v} = A^{k+1}\_{u\_1, ;} \\cdot A^{k+1}\_{;, v}$ appears to be problematic.

From a path interpretation perspective, the original statement suggests that the number of paths of length $k+1$ between nodes $u_1$ and $v$ can be derived by aggregating information from $A^{k+1}\_{u_1, ;}$, which accounts for all paths of length $k+1$ originating from $u_1$, and $A^{k+1}{;, v}$, which encompasses all paths of length $k+1$ terminating at $v$ from any other node. Would it not be more accurate to represent this relationship as $A^{k+1}\_{u_1,v} = A^{k}\_{u_1, ;} \cdot A^{1}\_{;, v}$?

Additionally, I would like to point out the presence of redundant statements, such as $A^{k+1}\_{u_1, v} \geq A^{k+1}\_{u_1, v}$ found in line 961, which could benefit from clarification or removal.

**Questions:**

1. Could you clarify what new insights are brought by the results in Table 1 compared to those in Figure 6 and Figure 3? The distinction between these results is not immediately clear to me.
2. In line 247, you mention that the sampling strategy should "(...) alter the graph representations in a way that is consistent with the target space." This aligns with the theoretical concerns in point 1. However, the experimental study performed pertains to counting substructures, a task where the information extracted by SC is expected to be relevant. How do you expect this sampling method to compare to other centralities in tasks where SC may not encode the most relevant information, such as network flow characterization?
3. The results under OSAN represent 1-OSAN?
4. Proposition 2 - Appendix B relates MPNNs with SE to DSS-GNN. The complete proposed method shows better results than 1-OSAN (assumed, see above) which is upper-bounded by the 2-WL, and better results than policy-learn which is expected to be more expressive than 1-WL but less than 4-WL. Where is the complete proposed method positioned in the WL framework for expressivity?
5. In line 424, it is stated that Figure 3 compares random sampling with SC. However, earlier (line 51), you state that random sampling is suboptimal. Why were other sampling strategies not tested to fully validate SC?
6. What was the method used to select the hyperparameter T? A brief explanation in the text would provide more clarity.
7. It was not addressed why a higher T, meaning more subgraphs, meaning more information leads to worse results. Is the information not useful? Is it connected to the sampling procedure not taking into account already sampled subgraphs? This is unclear to me.
8. For proposition 1, Step 2, the block matrix $W^{(j+1)\_0}$ seems to select the first $k+j$ columns not $k+j+1$? Consider $k=3$ for $j=1$. The block matrix will have dimensions $7 \times 7$, with only the block in the first spot of the diagonal performing a selection since the second block $I_{j-1}$ is omitted.
9. For the same proposition, I would like to seek clarification regarding the explanation provided for $AXW^{(j+1)_1}$. Specifically, the identity matrix is described as having dimensions $k - (j - 1)$, yet the reference appears to describe the selection of the last $j$ columns. Additionally, the process by which the iterations from $j = 1$ to $j = k$ contribute to the formation of the final vector is not entirely clear. I would greatly appreciate any further elaboration on these points to enhance my understanding.
10. Considering choices of T > 1 in the experiments, for theorem 4, what is the impact of k>1 for top-k Subgraph GNN compared to MPNN+CSE?

Minor:
1. The phrase starting in line 214, "If instrumental (...)",  is confusing, consider rewriting it.
2. Quartic is often misspelled as "quatric".
3. Line 890, the expression "1-hop neighborhood" is imprecise, I recommend 1-hop induced subgraph.
4. Line 836, misspelled "the" as "he".
5. Missing subscript on line 952?
6. No identification of what $m$ denotes in equation 15.

Suggestions (Optional):
1. Line 228, "(...) untrained 3-layer GIN". By untrained I assume the weights were initialized at random. If this is the case, for GIN, different random initialization should lead to some variance, even if small, in the results, leading to variance in the perturbations. It would be more robust to report the mean difference in perturbations across multiple random initializations, as this would account for variance in the results.
2. There is no direct comparison with some relevant network architectures like $I^2$-GNN that are not captured by 1-OSAN. I understand that $I^2$-GNN was used as a comparison point in MAG-GNN, but the results were quite close in the referred work, hence, I believe it would be useful to add such comparison. More importantly, works like ESC-GNN introduce a faster alternative to $I^2$-GNN. Since the presented work also focuses on efficiency, I believe a comparison with ESC-GNN would be interesting.
3. Since much of the code used is based on GraphGPS, the configs could be more carefully described to be easier to match the experiments in the paper.

The paper presents contributions that may be of limited novelty while having some inconsistencies. However, I am **very** open to revisiting and potentially increasing my evaluation score, provided the authors effectively address the identified weaknesses and respond to the questions posed. I encourage the authors to consider these aspects to enhance the manuscript's impact and clarity.

---

> ### Author Response · Authors · 2024-11-18
> **Response to Reviewer pVQU, pt1**
>
> We would like to deeply thank the reviewer for their outstanding effort. The review is comprehensive and constructive and we profoundly appreciate the provided inputs.
>
> Here in the following we specifically respond to the points they raised above.
>
> _**“Theoretical Concerns: The analysis based on perturbation is not sufficient to justify using the highest centrality nodes to guide the sampling procedure. Even though they will tend to produce the highest perturbation [...] their practical value in enhancing graph learning remains uncertain, there is no guarantee on the usefulness and attainability of such modifications.”**_
>
> Generally speaking, theoretical analyses can provide us with indications that a certain method could be practically beneficial, but, ultimately, experimental results will be critical in determining the practical value of a certain approach. As an example from adjacent literature: theoretical analyses on the ability of a GNN to discriminate (often exotic pairs of) non-isomorphic graphs do not give theoretical guarantees on generalization performance; experimental results, however, complement these analyses to demonstrate the utility of the method. In our work we followed a similar approach, in that we gathered helpful indications from theoretical and empirical observations and complemented these with experimental validations.
>
> More specifically, the theoretical observation reported in Section 3 takes the form of an upper-bound. It shows that the perturbation node marking induces w.r.t. the representation of a standard message-passing procedure is upper-bounded by a figure that is a function of the cumulative number of walks. What this implies is that marking nodes with the lowest such values may not guarantee to alter such representations enough. Aiming at conceiving an effective sampling strategy, we thus build upon this observation and complement it with experiments whose results are reported in Figure 1 and Table 1. There, we _empirically_ explore the behavior of perturbations when marking nodes with low and high values for (walk-based) centrality values (as well as random sampling strategies). First, the results in Figure 1 show that high-centrality marking tends to produce higher perturbations, something not necessarily entailed by the aforementioned bound-analysis. Second, the results in Table 1 further indicate that these higher perturbations better correlate with counts of various substructures.
>
> Complementary to Table 1 and Figure 1, we further explore how larger perturbations induced by higher-centrality marking can eventually be beneficial in enhancing the method’s performance after training. The results in Figure 3 and Appendix D already provide us with an indication of this, as marking nodes with highest centrality values performs better than random markings. We extended these experiments by also evaluating the performance of marking nodes with minimum centrality values. These results – expressed in Mean Absolute Error – further complement our findings:
>
> Triangles
> | Policy | 10 subgraphs | 20 subgraphs | 30 subgraphs | 40 subgraphs |
> |---|---|---|---|---|
> | Min Subgraph Centrality | 0.78 | 0.52 | 0.43 | 0.25 |
> | Random | 0.62 | 0.48 | 0.40 | 0.31 |
> | Max Subgraph Centrality | 0.20 | 0.10 | 0.03 | 0.03 |
>
> 4-cycles
> | Policy | 10 subgraphs | 20 subgraphs | 30 subgraphs | 40 subgraphs |
> |---|---|---|---|---|
> | Min Subgraph Centrality | 0.74 | 0.63 | 0.41 | 0.24 |
> | Random | 0.59 | 0.45 | 0.36 | 0.26 |
> | Max Subgraph Centrality | 0.38 | 0.12 | 0.08 | 0.04 |
>
> Taken together, all these theoretical and empirical observations provided us with an indication that marking nodes with higher centrality values represents a (simple and) valid sampling strategy. Let us conclude by remarking that the practical value of our approach is then eventually verified through the comprehensive set of experimental results on real-world datasets reported in Section 5.

---

> ### Author Response · Authors · 2024-11-18
> **Response to Reviewer pVQU, pt2**
>
> _**“The comparison between the adopted subgraph centrality, SC, and other centralities is not sufficient to prove SC's superiority [...] centrality measures often complement each other [...] Rather than claiming SC's superiority, it would be better to show in which tasks SC excels and acknowledge that other centrality measures may outperform it in different contexts.”**_
>
> It was not our intention to claim the absolute superiority of the Subgraph Centrality measure w.r.t. others, but we understand that our choice of experimenting with this measure in particular could have inadvertently conveyed this message.
>
> In general, we agree that different centrality measures may deliver more or less competitive performance in different contexts. Our experiments in Appendix D.1 constitute, in fact, a first inquiry into this possibility: we asked whether specific centralities excel more than others in counting certain substructures. These are, however, synthetic benchmarks and we acknowledge they may not necessarily give a satisfactory indication on real-world tasks.
>
> Accordingly, we ran additional experiments by comparing different centrality measures on the real-world peptides and molhiv datasets. We experimented in particular, with the Betweenness Centrality (BC), the Katz Index (KI) and the Subgraph Centrality (SC), obtaining the following results:
>
> | Centrality for Sampling | MolHIV | Peptides-Func | Peptides-Struct |
> |---|---|---|---|
> | BC | 78.86 (0.98) | 0.6749 (0.0066) | 0.2478 (0.0006) |
> | Katz | 79.58 (0.98) | 0.6756 (0.0056) | 0.2469 (0.0008) |
> | SC | 79.77 (0.70) | 0.6758 (0.0050) | 0.2466 (0.0010) |
>
> We see that the performances achieved by different measures are not dramatically different from each other, with those by the KI and SC being closer. In fact, centrality measures often exhibit a degree of correlation with each other, especially if from the same family, as it is the case of the walk-based KI and SC (see [1] and our Table 6). It is also worth noting that Subgraph Centrality can be more efficient to calculate than these other centrality measures using the networkx library. For example, for an Erdös-Renyi graph with 1000 nodes and p=0.5, we have
>
> | Centrality | Time (s) |
> |---|---|
> | BC | 83.12 |
> | Katz | 1.31 |
> | SC | 0.54 |
>
>
> Overall, we believe that specific centrality measures could work better than others depending on the task at hand, but, at the same time, our current ensemble of observations indicate that walk-based centrality measures – and, in particular, the Subgraph Centrality – offer the most competitive results for the lightest precomputation run-time. Given the additional support provided by the bound discussed in Observation 1, we think they constitute particularly strong candidates across use-cases.
>
> We would be happy to articulate this discussion in a new revision of the manuscript should the reviewer find it relevant.
>
> _**“RWSE vs CSE: While the proposed method (CSE) shows some advantages over RWSE, particularly from a sampling perspective as seen in Figure 7, its benefits appear limited. The experiments focus primarily on counting substructures, a relevant task but one that may not fully demonstrate CSE's broader applicability due to its inherent predisposition toward this type of task.”**_
>
> While we are grateful to the reviewer for bringing this point to our attention, we respectfully note that we do not see it as constituting a weakness of our work.
> First, we remark that our objective is to enhance the efficiency of Subgraph GNNs rather than to propose a new SE scheme that could surpass RWSEs in their generalization performance.
> Second, it is pivotal to highlight that, to the best of our knowledge, no previous work has suggested employing RWSEs for node ranking and/or subgraph sampling purposes. Thus, we do not think the relative superiority of one measure w.r.t. the other could anyhow impact the novelty, relevance or importance of our contribution.
> The focus on walk-based centralities arose naturally from our analysis in Section 2; it is then, by observing the resemblance between the terms involved in the calculation of these measures and the terms in RWSEs, that we believed a more comprehensive discussion would be informative for readers. This justified our articulated comparison in Appendix D.2.

---

> > ### Author Response · Authors · 2024-11-18
> > **Response to Reviewer pVQU, pt3**
> >
> > _**“Inconsistencies in Experiments: [...] the models used for comparison vary across datasets. It is understandable that in many cases the results for all datasets are not available [...] It would strengthen the results to ensure uniformity in model comparisons.”**_
> >
> > For each dataset, we chose the baselines that appeared to be the most relevant to answer our research questions (see the beginning of Section 5) in an informative manner: in Table 2 we compare to other sampling approaches, while Tables 3 and 4 focus more on comparing against strong dataset-specific baselines. We understand it would be preferred to have a completely uniform representation of all methods across all datasets, but this would require computational resources beyond those at our disposal during the time-frame of the present discussion period.
> >
> > However, in order to minimize the risk of perceiving our results as “cherry-picked”, we would be glad to report the results of baselines “Graph ViT” and “G-MLP-Mixer” not only on peptides datasets, but also on zinc and molhiv. These results have been added to Table 4 of the revision.
> >
> > _**“Missing Runtime Comparisons: [...] runtime comparisons are not provided for all datasets. Since computational efficiency is a key focus, these comparisons should be included in every experiment [...]”**_
> >
> > The wall-clock run-time analyses we provided are performed on the peptides datasets. We point out that, since these contain the largest-sized graphs across our benchmarking suite, these analyses will necessarily be the most informative.
> > We agree, however, that providing run-times on datasets with graphs from different distributions may increase completeness. We thus report additional timing results for zinc and molhiv, the largest datasets consisting of small molecular graphs.
> >
> > ZINC
> > | Model | Precompute (s) | Train (s/epoch) | Test (s) | Test MAE |
> > |---|---|---|---|---|
> > | GIN | 0.00 | 12.65 | 0.33 | 0.163 |
> > | HyMN (GIN, T=1) | 21.41 | 17.95 | 0.42 | 0.080 |
> > | GPS | 19.13 | 33.02 | 0.87 | 0.070 |
> >
> >
> > MOLHIV
> > | Model | Precompute (s) | Train (s/epoch) | Test (s) | ROC-AUC |
> > |---|---|---|---|---|
> > | GIN | 0.00 | 7.50 | 0.33 | 75.58 |
> > | HyMN (GIN, T=1) | 67.43 | 9.09 | 0.37 | 80.36 |
> > | GPS | 40.92 | 124.08 | 4.14 | 78.80 |
> >
> > We have populated these results in an additional timing comparison section in Appendix G in the latest revision
> >
> > _**“Confusing Proof: The presentation of Theorem 4 lacks clarity. [...]”**_
> >
> > After reviewing our proof, we agree some clarifications are needed in the presentation. Regrettably, the lack of headings and a proper paragraph formatting caused confusion.  More concretely, by “to achieve this” (line 912) we were referring to “showing that in both graphs the global node has the higher centrality”. We have fixed this in the new paper revision.
> >
> > _**“Furthermore, [...] the assertion in Equation 11 raises concerns regarding its mathematical validity and intuitive clarity.”**_
> >
> > With respect to equation 11, the reviewer was right to point out the equation was incorrect. This was a typo and the real equation should have been: $A^{k+1}_{u_2,v} = A^{k}_{u_2,:} \cdot  A_{:,v} \geq A^{k}_{u_2,:} \cdot  A_{:,u_1} = A^{k+1}_{u_2,u_1}$
> > exactly as the reviewer pointed out. _(We apologize but we are encountering troubles rendering the Latex formal here)_
> >
> > _**“Additionally, I would like to point out the presence of redundant statements, such as found in line 961, which could benefit from clarification or removal.”**_
> >
> > With respect to the redundancy in line 961, this is indeed a typo, it should say: $A^k_{v,u_1} \cdot A_{u_1, v} > A^k_{u_1,u_1} \cdot A_{u_1,u_1}$.
> >
> > We rewrote the proof of this lemma, corrected typos and added clarifications to better convey the proof strategy and avoid any source of confusion. We deeply thank the review for highlighting this aspect that allowed us to improve on the presentation of our work.
> >
> > We emphasize that the core idea of the proof remains unchanged, despite any potential confusion.
> >
> > _**“Could you clarify what new insights are brought by the results in Table 1 compared to those in Figure 6 and Figure 3? The distinction between these results is not immediately clear to me.”**_
> >
> > The results in Table 1 give us an indication that the perturbations induced by marking nodes with higher centralities (better) correlate with the count of various substructure counts. This provided us with another indication guiding the design of our sampling procedure.
> >
> > The results in Figures 3 and 6 report results obtained by _training_ Subgraph GNNs with various sampling strategies to predict the values of substructure counts. These results contributed to then validate our strategy when training the overall architecture to solve a prediction task (other than giving additional insights on the performance of other centrality measures).

---

> > > ### Author Response · Authors · 2024-11-18
> > > **Response to Reviewer pVQU, pt4**
> > >
> > > _**“[...] the experimental study performed pertains to counting substructures, a task where the information extracted by SC is expected to be relevant. How do you expect this sampling method to compare to other centralities in tasks where SC may not encode the most relevant information, such as network flow characterization?”**_
> > >
> > > As already mentioned above, it is possible that specific centrality measures may be more suited for sampling in certain tasks (similarly as to when specific SEs may be better suited for certain applications w.r.t. others). For example, one may reasonably expect that shortest-path-based centrality measures could perform well when estimating distances between nodes.
> > >
> > > However, it is important to remark the following. First, our Observation 1 suggests a connection between walk-based measures and the intrinsic behavior of message-passing, since it is possible to bound the impact of node marking in terms of the cumulative number of walks. Based on this connection, certain centrality measures could be better understood and further studied when used for subgraph sampling w.r.t. others. Second, the overall architecture also includes additional (learnable) components in its Subgraph GNN modules. As echoed by our expressiveness results in Section 4, these could extract information not necessarily encoded in the chosen centrality measure, contributing to making the whole method more flexible and generally adaptable.
> > >
> > > Overall, fine-grained results on these aspects may require more detailed analyses based on precise, specific assumptions, which we believe are interesting to investigate in follow-up research.
> > >
> > > _**“The results under OSAN represent 1-OSAN?”**_
> > >
> > > Yes, we confirm this.
> > >
> > > _**“[...] The complete proposed method shows better results than 1-OSAN (assumed, see above) which is upper-bounded by the 2-WL, and better results than policy-learn which is expected to be more expressive than 1-WL but less than 4-WL. Where is the complete proposed method positioned in the WL framework for expressivity?”**_
> > >
> > > It would be reasonable to hypothesize our full method is upper-bounded by 3-WL. A route towards this could proceed by showing that a DSS-GNN architecture can compute CSEs (we did, see Prop. 1) and can employ its internal components to approximate SC values from CSEs and nullify the contribution of subgraphs from nodes not included in the top-k centrality nodes. As DSS-GNN is upper-bounded by 3-WL [2] the upper-bound would then be inherited by HyMN.
> > >
> > > In any case, we importantly note that it would be necessary to carefully formalize these intuitions before asserting precise expressivity statements on our model.
> > >
> > > In regards to other methods, we could not find theoretical evidence PL is at least as expressive as 3-WL, which would give relevance to the reviewer’s claim that it is upper-bounded by 4-WL. Also, as for 1-OSAN, 2-WL is known to be equivalent to 1-WL; is the reviewer referring to the Folklore WL test?
> > >
> > > Let us finally remark that, in any case, although it can be informative to compare methods in their expressive power, more expressiveness does not causally induce better generalization properties.
> > >
> > > _**“In line 424, it is stated that Figure 3 compares random sampling with SC. However, earlier (line 51), you state that random sampling is suboptimal. Why were other sampling strategies not tested to fully validate SC?”**_
> > >
> > > Random sampling is, in fact, a particularly relevant baseline to us because it is efficient and it is not learnable (thus it does not require modifications to the training routine). We indeed tested several other sampling strategies of this kind in Appendix D: we included comparisons with strategies based on other centrality measures and even derived by the RWSE.
> > >
> > > _**“What was the method used to select the hyperparameter T? A brief explanation in the text would provide more clarity.”**_
> > >
> > > Hyperparameter T was selected to either match the choices made in other baselines, such as PL (i), or as the smallest possible value, i.e., T=1 (ii). Choice (i) contributed to making comparisons as informative as possible, (ii) is justified by our focus on efficiency. We have clarified this aspect in our next revision.

---

> > > > ### Author Response · Authors · 2024-11-18
> > > > **Response to Reviewer pVQU, pt5**
> > > >
> > > > _**“It was not addressed why a higher T, [...] leads to worse results. Is the information not useful? Is it connected to the sampling procedure not taking into account already sampled subgraphs?”**_
> > > >
> > > > This (interesting) behavior is also reported in other works, see, e.g., [3,4]. Although the reviewer’s intuition is reasonable, we signal that this phenomenon is observed also for methods which consider already sampled subgraphs, viz., PL. Because of this reason, more specific inquiries are required, and we believe they should be the object of future research endeavors.
> > > >
> > > > _**“For proposition 1, Step 2, the block matrix seems to select the first columns [...] ?”**_
> > > >
> > > > Thank you for your attention to detail. This is true, indeed the block matrix $W^{(j+1)_0}$ selects the first $k+j$ columns, and not the $k+j+1$ columns. We have edited the manuscript and addressed this issue – in red.
> > > >
> > > > _**“For the same proposition, I would like to seek clarification regarding the explanation provided for $AXW^{(j+1)_1}$. [...] Additionally, the process by which the iterations from j=1 to k contribute to the formation of the final vector is not entirely clear. I would greatly appreciate any further elaboration [...]”**_
> > > >
> > > > Thank you again for pointing this out. You are correct: the operation indeed selects the last $ k - (j - 1) $ columns of the matrix, not the last $ j $ columns as described in the paper. We have edited the manuscript and fixed this issue – in red.
> > > >
> > > > To clarify the structure of the proof: The proof follows an approach similar to induction, and can be summarized as follows:
> > > >
> > > > For a given $j$:
> > > > 1. We first fix the previously computed $ k + j $ slots, achieved through the operation $ XW^{(j+1)_0} $, and zero out the rest.
> > > > 2. At the same time we apply $ AXW^{(j+1)_1} $, which results in multiplying the rest of the columns of $X$, (the last $k - (j - 1)$ columns – which were zeroed out in (1) ) by $A$.
> > > > 3. Summing the results given by (1) and (2).
> > > > 4. Repeating (1), (2) and (3) for $j$ goes from $1 \rightarrow k$.
> > > >
> > > > By iterating steps (1), (2) and (3), and incrementing $j$, we gradually build the correct form of the final vector (as described in lines 846-847 of the paper), such that at the beginning of the $j$-th step, the first $k+j$ slots of the vector $\tilde{X}^{(k+1)}$ are correct, and matching the ones of the final output vector.
> > > >
> > > > _**“Considering choices of T > 1 in the experiments, for theorem 4, what is the impact of k>1 for top-k Subgraph GNN compared to MPNN+CSE?”**_
> > > >
> > > > Theorem 4 remains valid for k>1. We have extended the proof to the general case in the new version of the paper. Specifically, since our selection policy chooses the k nodes with the highest centrality, creating k disjoint copies of the graphs discussed in the proof of Theorem 4 provides a straightforward generalization for k>1. Additionally, as much of our work focuses on small bag sizes, the case of k=1 remains highly relevant.
> > > >
> > > > ---
> > > >
> > > > We finally share that we have implemented most of your "Minor" and noted your "Suggestions". Thank you!
> > > >
> > > > _References_
> > > >
> > > > [1] Estrada and Rodriguez-Velazquez. Subgraph centrality in complex networks, Physical Review E 2005
> > > >
> > > > [2] Frasca et al., Understanding and Extending Subgraph GNNs by Rethinking Their Symmetries, NeurIPS 2022
> > > >
> > > > [3] Bevilacqua et al. Efficient Subgraph GNNs by Learning Effective Selection Policies, ICLR 2024.
> > > >
> > > > [4] Bevilacqua et al. Equivariant Subgraph Aggregation Networks, ICLR 2022.

---

> > > > > ### Comment · Reviewer_pVQU · 2024-11-25
> > > > >
> > > > > I greatly appreciate the time and efforts the authors employed in answering the questions and even justifying the weaknesses pointed out. Their response shows receptiveness and commitment to improve their work.
> > > > >
> > > > > **Comparison of Centrality Measures and question 2**: The experiments mentioned in the answer (Centrality Sampling on MolHIV, Peptides, … and the time benchmarks) contribute to closing some of the concerns regarding the comparison between centralities. I still believe that this should be addressed more explicitly in the manuscript. Adding the mentioned tests and a statement similar to the one in “Overall, we believe that specific (...)” can clear most of the questions. Ultimately, even though centralities are often correlated when they have a common genesis, a reference in the manuscript to these concerns remains warranted. Centralities encapsulate distinct concepts when shaped by varying foundational influences (e.g., Krackhardt Graph [1]), and, moreover, there is no concrete theoretical guarantee that SC will consistently achieve optimality.
> > > > >
> > > > > **Inconsistencies in Experiments**: The addition of the new baselines “Graph ViT” and “G-MLP-Mixer” closes most of my main concerns.
> > > > >
> > > > > **Missing Runtime Comparisons**: While adding runtime to all experiments would be even more complete, I believe the justification given and the new data provided is enough to address the main concerns raised.
> > > > >
> > > > > **The WL Test**: I agree with the remarks made by the authors regarding the comparison made to the WL test. Even though adding a small section to the manuscript regarding WL comparison would elevate the work even more, I agree that such a topic requires a very careful and detailed formalisation. Given the scope of the proposed work, I believe it is not “mandatory” for such comparison to be added, being the informal discussion in the comments provided in the response enough for the interested reader.
> > > > > Regarding 1-OSAN, I apologise for the confusion, for this case, I was using the assumptions of the original work, meaning I was referring to the folklore or non-oblivious WL test.
> > > > >
> > > > > **In line 424, it is stated that Figure 3 (...)**: [Minor stuff] I understand the point of the authors, but I do not follow the reasoning to put Figure 3 in the main text and Figure 6 in the Appendix.
> > > > >
> > > > > **Hyperparameter T**: Apologies, but I could not find the clarification in the manuscript.
> > > > >
> > > > > **For the same proposition, I would like to seek clarification (...)**: I believe there is a typo in the manuscript. As acknowledged by the authors in the official response, the operation reflects the $k-(j-1)$ columns, not the $k-(j+1)$ as it is in the manuscript (line 850).
> > > > >
> > > > > Final comment:
> > > > > While certain highlighted aspects, such as the ones regarded in **theoretical concerns**, remain pertinent, I appreciate the authors’ perspective and acknowledge that these divergent opinions are not critical for the validity of the paper.
> > > > >
> > > > >
> > > > > Overall, while this work constitutes an incremental contribution, it is nonetheless grounded in a solid theoretical foundation. Furthermore, even though the empirical results are not very expressive when comparing to current state-of-the-art benchmarks, the methodology underlying these results achieves comparable performance through a distinct perspective. This alternative approach holds the potential to yield broader advantages (and raises interesting questions) within the field of efficient Subgraph GNNs. Hence, under the premise that the authors address the notes pointed out in this response I believe the work developed is enough to justify the rating “6: marginally above the acceptance threshold”. I will update my score.
> > > > >
> > > > >
> > > > > References:
> > > > > [1] Krackhardt, D. (1990). Assessing the Political Landscape: Structure, Cognition, and Power in Organizations. Administrative Science Quarterly, 35(2), 342–369. https://doi.org/10.2307/2393394

---

> ### Author Response · Authors · 2024-11-28
>
> Dear Reviewer pVQU,
>
> We hugely appreciate the time and effort you have made on our paper and believe that you have contributed greatly to the manuscript's improvement.
>
> **Comparison of Centrality Measures**
>
> We agree that the manuscript could be improved by better outlining our comparison between centrality measures. As you have suggested, we have added figure 6 to the main paper in order to show the advantages of centrality measures in general and, in particular, subgraph centrality. To further show this point, we have made a comment about their empirical advantages in lines #427-429 of the main paper. Here, we also reference the Appendix where we have performed a more extensive comparison between centrality measures and added the real-world experiments outlined in our initial rebuttal. We additionally provide remarks and indications about the choice of the centrality measure in accordance with our discussion (see lines #1286-1290)
>
> **Hyperparameter T**
>
> We are sorry for not being clearer about this parameter selection in the original text. In Appendix E.3 (lines #1516 - 1518), we have now specifically outlined the reasoning behind our choice of T on the different datasets in the manuscript. As discussed in the rebuttal, this was done to either match the choices made in other baselines, such as PL (i), or as the smallest possible value, i.e., T=1 (ii). Choice (i) contributed to making comparisons as informative as possible, (ii) is justified by our focus on efficiency.
>
> **Typo in proposition**
>
> Thanks for pointing this out! Indeed, we have changed this to $k − (j − 1)$ in the revision
>
> We would like to again thank the reviewer for the efforts they have made in reviewing and discussing our manuscript

---

### Official Review · Reviewer_8UxD · 2024-11-01

**Soundness:** 3
**Presentation:** 2
**Contribution:** 3
**Rating:** 6
**Confidence:** 3

**Summary:**

This paper introduces a novel method called Hybrid Marked Network (HyMN), aimed at balancing
computational efficiency and model expressiveness. The approach combines subgraph GNNs
with structure encodings (SEs) based on walk centrality measures. The main innovation lies in
utilizing subgraph centrality for subgraph sampling and structure encoding, allowing for the
maintenance of prediction accuracy while reducing the required number of subgraphs, thus
lowering computational costs while preserving the model's expressive capacity. Experimental
validation on synthetic and real world tasks demonstrates that HyMN outperforms other state-of
the-art GNN methods while reducing computational demands

**Strengths:**

1. The walk centrality-based subgraph selection method achieves efficient subgraph sampling,
simplifying the model while ensuring performance, making it suitable for larger subgraphs.
 2. Experiments on various tasks showcase HyMN's adaptability and performance.
 3. By ranking nodes based on their centrality values, and mark the top-scoring ones, the model
reduces computation time, making it applicable to a wider spectrum of downstream tasks.

**Weaknesses:**

Abstract and Background Information:
The abstract is incomplete, as it lacks essential background information. This omission leaves me confused about the specific problem the paper aims to address. A well-defined problem statement is crucial for understanding the motivation behind the research and its significance.

Experimental Validation of CSEs:
I noticed that experiments on the effect of CSEs in the Peptides and Zinc datasets were not conducted. This absence raises questions about the impact of incorporating CSEs (Q4). Clarifying this point is essential for understanding how CSEs contribute to the overall findings of the study. Consider including experimental results or justifications for their omission.

Exploration of Selection Strategies:
The paper would benefit from exploring more complex selection strategies and different walk-based centrality measures. This exploration could provide deeper insights into the dynamics at play and strengthen the overall analysis.

Typos and Formatting Issues:
While typos and formatting are not the most critical issues, there are several areas that require attention. For instance, the indentation of the abstract does not align with the template requirements. Additionally, line 194 contains a grammatical error with "is are" used simultaneously, which should be corrected. Some equations lack punctuation at their end, where it is needed, leading to inconsistencies. Lines 206-210 stray from the scope of Definition 1 and should not be italicized. Furthermore, Algorithm 1 lacks a title, which is necessary for clarity and organization.

Section 2 - Clarification of Content:
Although Section 2 is titled “PRELIMINARIES AND RELATED WORK,” it primarily focuses on related work without adequately presenting the foundational definitions necessary for understanding the subsequent content. It would be beneficial to include a clearer explanation of the definitions and concepts that readers should be familiar with before delving into the related literature.

**Questions:**

Refer to Weakness.

---

> ### Author Response · Authors · 2024-11-18
> **Response to Reviewer 8UxD**
>
> We thank the reviewer for their feedback. Here, we respond to the weaknesses and questions point by point.
>
> _**“Incomplete abstract? [...]”**_
>
> In order to add further contextualization, and in accordance with the reviewer’s suggestion, we propose to rephrase the first sentences of the abstract as follows:
>
> _Popular extensions of Graph Neural Networks (GNNs) are Subgraph GNNs, which enhance expressiveness by processing bags of subgraphs obtained from predefined selection policies. Although more powerful, these networks suffer from a problematic computational complexity. Addressing this issue, we propose an expressive and efficient approach which leverages walk-based centrality measures, both as an effective subgraph sampling strategy for Subgraph GNNs and as a powerful form of Structural Encoding (SE). [...]_
>
> We hope that this new version improves clarity in regards to the addressed problem, but we are open to further changes should the reviewer have additional comments/suggestions.
>
> _**“Experimental Validation of CSEs: I noticed that experiments on the effect of CSEs in the Peptides and Zinc datasets were not conducted.”**_
>
> We thank the reviewer for suggesting important comparisons which can improve the paper. We have now run some further experiments to appreciate the effect of CSEs on Zinc and Peptides, see below:
>
> | Model | Zinc | Peptides-Func | Peptides-Struct |
> |---|---|---|---|
> | GIN | 0.163 (0.004) | 0.6558 (0.0068) | 0.2497 (0.0012) |
> | GIN+CSE | 0.092 (0.002) | 0.6619 (0.0077) | 0.2479 (0.0011) |
> | HyMN (GIN, T=1) w/out CSE | 0.125 (0.004) | 0.6758 (0.0050) | 0.2466 (0.0010) |
> | HyMN (GIN, T=1) | 0.080 (0.003) | 0.6857 (0.0055) | 0.2464 (0.0013) |
>
> These results further show that adding even one subgraph with our approach can be beneficial and that additionally using the centrality measure as a structural encoding can also improve performance. This is in line with what is argued in the paper and we hope these further experiments provide additional clarity to the manuscript. We have added these results to the manuscript in Appendix D.3 and reserve to further extend these experiments to larger values of $T$.
>
> _**“Exploration of selection strategies: the paper would benefit from exploring more complex selection strategies and different walk-based centrality measures.”**_
>
> Our main goal is to develop an effective, simple and, importantly, _efficient_ sampling strategy, as opposed to more complex techniques requiring lengthy and more sophisticated training protocols. Given our focus on efficiency, in our paper we further explored the impact of strategies based on different centrality measures or based on Random Walk Structural Encodings. All these results can be found in Appendix D.
>
> We agree however that more experiments on real-world datasets and by other walk-based centrality measures could be informative and improve our manuscript.
> Here we report additional results obtained by the Betweenness Centrality (BC), the Katz Index (KI) and the Subgraph Centrality (SC), on the peptides and molhiv datasets. Both KI and SC are walk-based centrality measures.
>
> | Centrality for Sampling | MolHIV | Peptides-Func | Peptides-Struct |
> |---|---|---|---|
> | BC | 78.86 (0.98) | 0.6749 (0.0066) | 0.2478 (0.0006) |
> | Katz | 79.58 (0.98) | 0.6756 (0.0056) | 0.2469 (0.0008) |
> | SC | 79.77 (0.70) | 0.6758 (0.0050) | 0.2466 (0.0010) |
>
> We observe that, even on these real-world tasks, the two walk-based centrality measures tend to outperform the BC, while we still could not find empirical evidence of a measure outperforming SC. These results are in line with those in Appendix D and provide further indication that walk-based approaches could be more suited to subgraph sampling when combined with message-passing backbones.
>
> _**“Typos and formatting issues”**_
>
> Thank you for pointing this out. We have made our best to improve on this aspect: our paper revision fixes typos and related issues in order to improve clarity (see corrections in red).
>
> _**“Section 2: clarification of content – though Section 2 is titled “PRELIMINARIES AND RELATED WORK,” it primarily focuses on related work without adequately presenting the foundational definitions”**_
>
> In agreement with the reviewer, in our next revision: (i) we change the title of Section 2 to “Related Work” and include a new  supplementary Appendix F, titled “More on Subgraph GNNs and their Complexity”. This appendix introduces Subgraph GNNs and discusses in detail aspects related to our present contribution: node marking policies, computational complexity aspects and the approach of subgraph sampling.

---

> > ### Author Response · Authors · 2024-11-28
> >
> > Dear Reviewer 8UxD,
> >
> > Thank you very much for the time you have dedicated to reviewing our paper.
> >
> > We believe that our new section and changes have resolved your concerns with regards to the background information and the clarification of content. We have also run the CSE and centrality comparison experiments which you suggested, other than fixing the formatting, this contributing to enhance the quality of our paper.
> >
> > Please let us know if any questions remain unaddressed, and if you would be open to reconsider your current evaluation of our submission.
> >
> > Thank you!

---

> > > ### Comment · Reviewer_8UxD · 2024-12-02
> > >
> > > Dear Authors,
> > >
> > > Thank you for your detailed response. However, I noticed that many issues I raised in my initial comments remain unresolved in the updated version of your paper. Setting aside more complex concerns, one of the most apparent is that several formatting issues I explicitly pointed out have not been addressed. For example:
> > >
> > > - Some equations include punctuation (e.g., Equation 4), while others do not (e.g., Equations 1, 2, and 3), which need to be consistent.
> > > - Lines 206–210 stray from the scope of Definition 1 and should not be italicized.
> > > - The indentation in the abstract is incorrect.
> > >
> > > These, among other issues, were highlighted in my initial comments but remain unchanged in the revised manuscript.
> > >
> > > If you believe any of my comments were incorrect, I would have appreciated clarification. However, your response stated that these issues were addressed, which is evidently not the case. This discrepancy is disappointing, as it gives the impression that my feedback was not taken seriously despite the time and effort I invested in reviewing your work.
> > >
> > > That said, I will maintain my original score, as my overall evaluation of your work remains unchanged.
> > >
> > > Reviewer 8UxD

---

> > > > ### Author Response · Authors · 2024-12-02
> > > > **Further clarifying response to Reviewer 8UxD**
> > > >
> > > > Dear Reviewer 8UxD,
> > > >
> > > > Thank you for you response. We indeed really do appreciate the time you spent reviewing the manuscript and we did take your feedback very seriously and tried to resolve all of the concerns.
> > > >
> > > > **Abstract and Background Information**
> > > >
> > > > We proposed a new abstract and we also added a new background section in Appendix F.
> > > >
> > > > **Experimental Validation of CSEs and Exploration of Selection Strategies**
> > > >
> > > > We ran additional experiments on CSEs and comparisons to other centrality measures.
> > > >
> > > > **Typos and Formatting Issues**
> > > >
> > > > We have indeed fixed the typos we could find in the manuscript, and we are sorry for not clarifying the changes made, and our position with respect to the raised formatting issues. Regarding your explicit concerns:
> > > > - Line 194 "is are" used simultaneously: "$\phi^{(l)}$'s are update functions and $\phi^{(L+1)}$ is". These are referring to different subjects, of which one is plural and the other is singular, so we believed that "are" and "is" were used correctly;
> > > > - We do not have punctuation for Equations 1, 2 and 3 as they are embedded as part of the sentence. We instead have a full-stop after Equation 4 as it is the end of the sentence. We thus judged our formatting coherent.
> > > > - Lines 206-210 are enclosed within an "Observation", and are indeed, the most crucial component thereof. This is the reason why we kept them italicized.
> > > > - "Algorithm 1 lacks a title": We had actually titled it originally as "Hybrid Marking Network" and this is still kept in the current revision.
> > > >
> > > > Nonetheless, we agree with the reviewer that we have unintentionally overlooked the issue regarding the abstract indentation. We remark it was not our intention to ignore such a comment – we just inadvertently failed to notice this, while focusing on addressing the main abstract concern ("incomplete abstract").
> > > >
> > > > Thanking the reviewer for pointing this out again, we have now fixed this minor issue, which was apparently due to the clashing package "compactenum". By replacing this with the equivalent "enumitem", we managed to completely fix the problem. Regrettably, we are not allowed to upload a new revision at this time, but we are happy to report the manuscript is now fully compliant with the expected formatting style.
> > > >
> > > > **Section 2 - Clarification of Content**
> > > >
> > > > We remark we have added a new, comprehensive background section in Appendix F.
> > > >
> > > > ----
> > > >
> > > > We understand that the lack of elaboration on the typos has caused concern and it was not our intention to undervalue your review. Regardless, the reviewer stated the above are only the most apparent concerns, and that "more complex" ones are set aside – we would greatly appreciate if you could elaborate more on these, to help us improve the manuscript further.

---

### Official Review · Reviewer_i9ZD · 2024-11-02

**Soundness:** 3
**Presentation:** 2
**Contribution:** 2
**Rating:** 5
**Confidence:** 3

**Summary:**

To balance efficiency and expressiveness in subgraph GNNs, this paper proposes a novel framework that utilizes walk-based centrality measures for subgraph subsampling and structural encoding. The necessity of using centrality measures is demonstrated through theoretical analysis and experimental validation. Experimental results show the effectiveness of HyMN across various tasks and datasets.

**Strengths:**

1. The paper proposes a novel framework that balances the efficiency and expressiveness of subgraph GNNs.
2. This paper provides a comprehensive theoretical analysis and experimental validation of why the simple subgraph centrality measure is needed for subgraph GNN.
3. The experiment results demonstrate the effectiveness and efficiency of the proposed method.

**Weaknesses:**

1. The two main components (subsample subgraph using centrality measure, and centrality-based structural encoding) are similar with previous work.[1,2]
2. A more detailed introduction to the key background of this work would be helpful(i.e. the background and method of the node marking)
3. More backbone(except GIN) and more baseline should be preferred to be considered by authors.

[1] Sun, Qingyun, et al. "Sugar: Subgraph neural network with reinforcement pooling and self-supervised mutual information mechanism." *Proceedings of the web conference 2021*. 2021.

[2] Rampášek, Ladislav, et al. "Recipe for a general, powerful, scalable graph transformer." *Advances in Neural Information Processing Systems* 35 (2022): 14501-14515.

**Questions:**

1. Could the author provide more evidence to support the validity of the two claims in line 182?

---

> ### Author Response · Authors · 2024-11-18
> **Response to Reviewer i9ZD**
>
> We appreciate feedback and response received by the reviewer. Below, we specifically address the concerns and questions raised.
>
> _**“The two main components (subsample subgraph using centrality measure, and centrality-based structural encoding) are similar with previous work.”**_
>
> Our contribution is, in fact, orthogonal to these works. Indeed, in [1], the authors use Reinforcement Learning to pool representations of subgraphs of the original graph. In GraphGPS [2], the authors propose design strategies for Graph Transformers (GTs), including the use of a random-walk based structural encoding. Effectively, neither of these works consider walk-based centralities and neither of these works subsample subgraphs tied to specific nodes in the original graph to improve expressivity in an efficient way.
> While these papers may discuss somewhat related ideas, our work substantially differs both in its technical contributions and in the problem it addresses, and we believe it clearly provides novel inputs and methodologies to the community.
> Either way, we have added [1] as a citation to improve completeness in our new background section in the Appendix which we discuss below.
>
> _**“A more detailed introduction to the key background of this work would be helpful”**_
>
> We agree with the reviewer that adding a more detailed background would improve the quality of our manuscript. We include such an extended discussion in a new supplementary Appendix F, namely: “More on Subgraph GNNs and their Complexity”. This section introduces Subgraph GNNs in general and discusses, more in detail, those aspects which pertain more to our present contribution: node marking policies, computational complexity aspects and the approach of subgraph sampling.
>
> _**“More backbone (except GIN) and more baselines should be considered”**_
>
> We report that, beyond GIN, we have already experimented with GCN as the backbone on the Peptides datasets, which proved very successful. GCN and GIN are not only the two most popular MPNN architectures, but also, the most widely adopted Subgraph GNN backbones. Due to this, we believe our experiments are particularly informative for the community.
> In any case, in agreement with the reviewer’s suggestion, we have performed further comparisons that can help improve the quality of our manuscript. In particular, we have additionally run GCN as a backbone on OGB molecular datasets, with results reported below.
>
> | Model | MolHIV | MolBace | MolTox |
> |---|---|---|---|
> | GCN | 76.06 (0.97) | 79.15 (1.44) | 75.29 (0.69) |
> | HyMN (GCN, T=2) | 76.82 (1.20) | 83.21 (0.51) | 76.07 (0.50) |
>
> Interestingly, we find that a GCN backbone achieves the best results on MOLBACE. We will ensure that all of the substantial comparisons with GIN and GCN as backbones will be added to the paper.
> We would like to further report that we have run experiments with yet another backbone: GatedGCN [3]. This complements backbones used in the Graph Transformer literature such as GPS. The results on Peptides and MolHIV are shown below.
>
> | Model | MolHIV | Peptides-Func | Peptides-Struct |
> |---|---|---|---|
> | GatedGCN | 78.27 (0.65) | 0.6558 (0.0068) | 0.2497 (0.0007) |
> | GatedGCN + CSE | 80.33 (0.56) | 0.6611 (0.0058) | 0.2482 (0.0010) |
> | HyMN (GatedGCN, T=2) w/out CSE | 79.02 (0.91) | 0.6723 (0.0079) | 0.2473 (0.0013) |
> | HyMN (GatedGCN, T=2) | 81.07 (0.21) | 0.6788 (0.0052) | 0.2471 (0.0007) |
>
> Again, these results empirically strengthen the findings already reported in the paper. The walk-based centrality structural encoding can improve performance over the backbone MPNN, and we can improve the performance even further by using it for sampling subgraphs.
>
> As for the baselines, we selected those that were either directly comparable to our approach (Policy-Learn, Full-bag) or popular, prototypical SOTA methods such as Graph Transformers (GPS) or higher-order methods (CIN). Our method clearly demonstrated its utility across all experiments and against both categories of approaches as it either outperformed or worked equally well with reduced run-time.
>
> Either way, to improve on uniformity and completeness, we have added two additional baselines on the Zinc and MolHIV datasets: “Graph ViT” and “G-MLP-Mixer” which can be seen in Table 4 of the revision.

---

> > ### Author Response · Authors · 2024-11-18
> > **Response to Reviewer i9ZD, pt 2**
> >
> > _**“Could the author provide more evidence to support the validity of the two claims in line 182?”**_
> >
> > We thank the reviewer for asking this. The claim in line 182 has to be interpreted as a guiding intuition leading the design of our subgraph sampling strategy. This intuition assisted us in conceiving our method and was eventually practically validated in our experiments.
> >
> > Essentially, if we are to pick a small number of subgraphs to approach the performance of a full-bag method, we would like to avoid selecting those that may not provide significant additional information w.r.t. what a standard message-passing procedure could compute – hence the intuition on studying perturbations. At the same time, one would hope that this additional information is as meaningful for the task at hand as possible. As for this last point, one could reason in terms of graph discrimination. Ideally, it would be desirable to separate two 1-WL equivalent graphs when they are associated with different labels – in this sense, our desideratum is not only that their representations are perturbed enough, but also that they are different when associated with different learning targets.
> >
> > ---
> > Thanks again for your review and the comments which have helped improve the manuscript. We hope that we have sufficiently addressed your points and concerns. Please let us know if this remains unclear or if you consider any other discussion points to be open.
> >
> > _References_
> >
> > [1] Sun et al. Sugar: Subgraph neural network with reinforcement pooling and self-supervised mutual information mechanism. Proceedings of the web conference 2021.
> >
> > [2] Rampášek et al. Recipe for a General, Powerful, Scalable Graph Transformer. NeurIPS 2022.
> >
> > [3] Bresson and Laurent. Residual Gated Graph ConvNets, arXiv preprint 2018

---

> > > ### Author Response · Authors · 2024-11-28
> > >
> > > Dear Reviewer i9ZD,
> > >
> > > Thank you very much for the time you have dedicated to reviewing our paper.
> > >
> > > We believe that our new section and changes have resolved your concerns with regards to the background information and relation to previous works. We have also run the additional backbone experiments which we feel has strengthened the message of the manuscript.
> > >
> > > Please let us know if any questions remain unaddressed, and if you would be open to reconsider your current evaluation of our submission.
> > >
> > > Thank you!

---

### Official Review · Reviewer_bWnw · 2024-11-02

**Soundness:** 3
**Presentation:** 3
**Contribution:** 3
**Rating:** 6
**Confidence:** 3

**Summary:**

The paper proposes a model termed Hybrid Marketing Network (HyMN), which is designed based on two intuitions which may potentially improve the GNN performance. First, marking the nodes included in Subgraph GNNs using walk-based subgraph centrality measures as an efficient strategy. Second, augmenting the node features with the same centrality measures as Structureal Encodings (SEs).  The key insight is that walk-based centrality measures serve both as effective indicators for subgraph importance and as infrmative structrual features. The authors theoretically analysed the node marking strategy with graph perturbation theory and demonstrate that their approach effectively balance expressiveness and efficiency.

**Strengths:**

1. the theoretical analysis provide a solid foundation for their proposed method. The utilization of perturbation theory and expressive theory looks correct for me.
2. This method is neat in design, which only requires minimal computation overhead compared with baselines while maintaining competitive performance.
3. The experimental validation is comprehensive, besides comparing their method with SOTAs, they conduct synthetic experiments for counting substructures, detailed ablations and experiment time analysis.

**Weaknesses:**

1. The sampling strategy does not consider interactions between sampled subrgaphs that are already sampled, which can lead to potential redundancy.
2. The analysis focus primarily on single-node marking. While mentioned in the limitations, extending the analysis to multi-node marking could provide addiitonal insights. Imagine a social network representing employees in a company. In this network, there's a team of three senior managers (say Alice, Bob, and Carol) who work closely together and have very similar connections to other employees. They all interact with the same team members, attend the same meetings, and collaborate on similar projects. According to your approach, since all three managers have high centrality scores due to their positions, the algorithm might select both Alice and Bob for marking. However, because their network positions and connections are so similar, marking both of them provides largely redundant information. It would be more informative to mark Alice (representing the management team's perspective) and then perhaps mark someone from a different department or organizational level, like a developer or a project coordinator, to capture a different aspect of the company's structure.
3. The approach can be less effective on graphs where walk-based centrality measures don't align with task objectives. Consider a drug discovery task where we're trying to predict whether molecules will bind to a specific protein receptor. The binding activity often depends on specific functional groups located at the periphery of the molecule. Take acetylsalicylic acid (aspirin) as an example. The molecule's binding properties are largely determined by its acetyl group at the edge of the molecule, but the walk-based centrality measure would give more importance to the central benzene ring because it participates in more walks through the molecular graph. In this case, marking nodes based on centrality would emphasize the structurally central but functionally less relevant parts of the molecule, while potentially overlooking the peripheral functional groups that actually determine the binding behavior. This mismatch between structural centrality and functional importance could lead to suboptimal performance on specific prediction tasks.

**Questions:**

1. Have you investigated how the performance of you sampling strategy varies with graph density? Intuitively, in very sparse graphs, walk-based centrality might be less informative.
2. In the perturbation analysis, could the bound be tightened by considering specific graph properties or structures? For instance, does the bound become tighter for trees or graphs with bounded degree?
3. Is there any strategies to extend the perturbation analysis to handle edge features or different types of marking strategies?

---

> ### Author Response · Authors · 2024-11-18
> **Response to Reviewer bWnw**
>
> We thank the reviewer for their feedback and their positive response. Here, we respond to the weaknesses and questions point by point.
>
> _**“The sampling strategy does not consider interactions between sampled subgraphs that are already sampled, which can lead to potential redundancy. The analysis focus primarily on single-node marking. While mentioned in the limitations, extending the analysis to multi-node marking could provide addiitonal [sic.] insights.”**_
>
> These are very important and interesting points which you make. Indeed, future work should consider the effect of incorporating information from previously sampled subgraphs, similar to Policy-Learn [1]. We also note that the example provided by the reviewer to highlight the importance of multi-node marking (“Alice, Bob, Carol”) could also be addressed by considering interactions with already sampled subgraphs; the sampling strategy could, for example, refrain from marking nodes with small shortest-path-distances w.r.t. already marked nodes.
>
> Here we note that, when empirically comparing against Policy-Learn on a small number of subgraphs, our simpler strategy can match and sometimes outperform this method. This leads us to hypothesize the aforementioned interactions may be less practically impactful when considering very small bags. It is also important to remark that more sophisticated strategies accounting for interactions with already sampled subgraphs would likely lead to more complex and less computationally efficient approaches, against our main high-level desiderata.
>
> We hope that, in any case, this paper and the connection provided to perturbation analysis provides a “solid foundation” which can be extended in future work to more sophisticated sampling strategies which could also consider concurrently marking multiple nodes.
>
> _**“The approach can be less effective on graphs where walk-based centrality measures don’t align with task objectives. [...]”**_
>
> This is another relevant observation and is, in fact, a difficulty encountered by all graph learning architectures which use “pre-defined features”, e.g. Structural Encodings. However, we do find notable benefits with our approach. It is a simple and efficient measure that we observed can match or outperform learnable policies (Table 2). In Tables 3 and 4, we found our method to work well when compared against many strong baselines on a set of experiments you also positively highlighted to be comprehensive.
>
> In reference to the provided drug-discovery example, we note that, although the acetyl group may not necessarily be the object of marking, its representation can in any case be altered and potentially be improved by marking other more central molecular sites, provided enough message-passing layers.
>
> Overall, we found that the walk-based centralities are fast to precompute, arise from our perturbation analysis and perform well on a range of tasks, indicating that they are strong candidates for our model. However, we note that more specific measures for particular applications could be explored.
>
> _**“Have you investigated how the performance of your sampling strategy varies with graph density?”**_
>
> The real-world datasets we used in the paper are all sparse. Zinc (mean #nodes: 23.2 , mean #edges: 24.9), MOLHIV (mean #nodes: 25.5 , mean #edges: 27.5), Peptides (mean #nodes: 150.9 , mean #edges: 307.3) all demonstrate the performance improvements of using our walk-based centrality on sparse graphs. Additionally, our counting substructure experiments are performed on much denser graphs (mean #nodes: 60 , mean #edges: 240) giving us evidence that our approach can work well across graph densities.
>
> _**“In the perturbation analysis, could the bound be tightened by considering specific graph properties or structures? For instance, does the bound become tighter for trees or graphs with bounded degree?”**_
>
> We do not clearly see an obvious way to improve the tightness of the upper-bound based on characteristics of the graph distribution at hand. However, it could be interesting to derive more specific – and perhaps tighter – upper-bounds for recent GNN architectures that are tailored to specific graph distributions such as planar graphs [2][3].

---

> > ### Author Response · Authors · 2024-11-18
> > **Response to Reviewer bWnw, pt 2**
> >
> > _**“Is there any strategies to extend the perturbation analysis to handle edge features or different types of marking strategies?”**_
> >
> > The work in [4] also explores perturbations caused by removing nodes and edges, so our analysis would be easily extended to Subgraph GNNs equipped with node and edge-deletion selection policies. Accounting for edge-features could be less immediate. As the results on node perturbations are obtained by leveraging the structure of computational trees, one possibility could be to define computational trees also accounting for the inclusion of edge features in the message-passing computations and then work-out bounds with strategies similar to the ones in [4].
> >
> > ---
> >
> > We hope that we have sufficiently addressed your points and concerns. Please let us know if this remains unclear or if you consider any other discussion points to be open.
> >
> > _References_
> >
> > [1] Bevilacqua et al. Efficient Subgraph GNNs by Learning Effective Selection Policies. ICLR 2024.
> >
> > [2] Dimitrov et al. PLANE: Representation Learning over Planar Graphs. NeurIPS 2023.
> >
> > [3] Bause et al. Maximally Expressive GNNs for Outerplanar Graphs. NeurIPS GLFrontiers 2023.
> >
> > [4] Chuang et al. Tree Mover’s Distance: Bridging Graph Metrics and Stability of Graph Neural Networks. NeurIPS 2022.

---

### Author Response · Authors · 2024-11-18
**General response**

We are very grateful to all reviewers for their comprehensive responses which, we gladly note, have highlighted several positive traits of our presented work.

The proposed framework has been found _“neat in design”_ (bWnw), _“novel”_ (i9ZD), _adaptable_ (8UxD) and to be addressing _“an issue [...] that is extremely relevant for the community”_ (pVQU).

Reviewers have appreciated our provided theoretical analyses, which they believe provide _“a solid foundation”_ for our method (bWnw), are _“comprehensive”_ (i9ZD), _“well founded”_, _“creative”_ and _well motivate the design decisions_ (pVQU).

Reviewers bWnw, i9ZD and 8UxD commend the _simplicity of the solution_ and its ability to attain _competitive performance with computational efficiency_. They have praised the _comprehensiveness_ of our experimental validation, with its _“detailed ablations and experiment time analysis”_ (bWnw).

The reviewers have also kindly provided interesting points of discussion and raised relevant questions which we address specifically below.

We finally remark that, following the reviewers’ suggestions we have run additional experiments:
1. Additional run-time analyses on zinc and molhiv;
2. Additional experiments on Subgraph GNN backbones;
3. Experiments on real-world datasets with other centrality measures, including timing their precomputation;
4. Min-centrality marking on substructure counting;
5. Additional ablations on the impact of CSEs on the Peptides and ZINC datasets.

Overall, the above confirm that:
- Our approach is efficient both in its precomputation and forward pass (1., 3.,), is robust to the choice of the backbone architecture (2.);
- Sampling via (larger values of) walk-based centrality measures and, in particular, the Subgraph Centrality is a robust strategy across benchmarks (3., 4.)
- Both marking and (centrality-based) structural encodings provide benefits when used jointly over using either in isolation (5.)

A new revision has been posted. It includes improvements based on reviewers’ comments as well as salient additional results from the above experiments.

---

### Comment · Area_Chair_ikx2 · 2024-11-23
**Reminder: Please Review Author Responses**

Dear Reviewers,

As the discussion period is coming to a close, please take a moment to review the authors’ responses if you haven’t done so already. Even if you decide not to update your evaluation, kindly confirm that you have reviewed the responses and that they do not change your assessment.

Thank you for your time and effort!

Best regards, AC

---

### Meta-Review · Area_Chair_ikx2 · 2024-12-07

**Metareview:**

This paper introduces a method aimed at improving the efficiency of Subgraph GNNs. The approach leverages walk-based subgraph centrality measures to guide subgraph sampling and proposes using centralities that may highlight structurally sensitive nodes as the centers for subgraph sampling. Experimentally, the method demonstrates reasonably good performance compared to baseline methods across various graph-related tasks.

The main strength of this work lies in its efficiency improvements for Subgraph GNNs while maintaining competitive performance on multiple datasets. The key concern is the limited novelty, as the proposed method is primarily a heuristic for incrementally improving existing subgraph GNN pipelines. It remains unclear whether these heuristics are applicable to a broader domain or some critical applications, such as drug discovery, where the most relevant nodes may correspond to important functional groups rather than structurally sensitive ones, as mentioned by two reviewers. After reading the paper by myself, I would like to raise a minor question: As the goal is to have more expressive yet computationally efficient graph learning, the authors may consider comparing with other methods that directly incorporate other structural features into GNNs or adopt faster architectures such as those in [1,2]. These methods seem to have shown practical efficiency and strong performance. The authors are suggested to compare with these efficient and expressive pipelines to demonstrate the necessity of accelerating subgraph GNNs.

Overall, while this paper achieves some valid findings, I consider it a borderline submission. Given the high bar for ICLR, I lean toward recommending rejection.

[1] Recurrent Distance-Encoding Neural Networks for Graph Representation Learning, Ding et al., ICML 2024
[2] What Can We Learn from State Space Models for Machine Learning on Graphs? Huang et al., arxiv 2024

**Additional Comments On Reviewer Discussion:**

The key concern revolves around the empirical comparisons, such as missing baselines and computational evaluations, which the authors have addressed effectively in their response. However, the concerns influencing my decision are: Raised by Reviewer bWnw and Reviewer pVQU, regarding whether the proposed heuristic for selecting nodes can be effectively applied to other critical applications; Raised by Reviewer pVQU, about the incremental nature of the contributions.

Although both reviewers ultimately leaned toward borderline acceptance, these concerns are somewhat fundamental.

---

### Decision · Program_Chairs · 2025-01-22

Reject